# Negative differential resistance as a critical indicator for the discharge capacity of lithium-oxygen batteries

Yoko Hase [1], Yasuhiro Komori[2], Takayoshi Kusumoto[2], Takashi Harada [3], Juntaro Seki[1], Tohru Shiga [1], Kazuhide Kamiya [2,3] & Shuji Nakanishi [2,3]

In non-aqueous lithium-oxygen batteries, the one-electron reduction of oxygen and subsequent lithium oxide formation both occur during discharge. This lithium oxide can be converted to insulating lithium peroxide via two different pathways: a second reduction at the cathode surface or disproportionation in solution. The latter process is known to be advantageous with regard to increasing the discharge capacity and is promoted by a high donor number electrolyte because of the stability of lithium oxide in media of this type. Herein, we report that the cathodic oxygen reduction reaction during discharge typically exhibits negative differential resistance. Importantly, the magnitude of negative differential resistance, which varies with the system component, and the position of the cathode potential relative to the negative differential resistance determined the reaction pathway and the discharge capacity. This result implies that the stability of lithium oxide on the cathode also contributes to the determination of the reaction pathway.

[1] Toyota Central R&D Labs., Inc., 41-1 Yokomichi, Nagakute, Aichi 480-1192, Japan. [2] Research Center for Solar Energy Chemistry, Osaka University, 1-3 Machikaneyama, Toyonaka, Osaka 560-8531, Japan. [3] Department of Chemistry, Graduate School of Engineering Science, Osaka University, 1-3 Machikaneyama, Toyonaka, Osaka 560-8531, Japan. Correspondence and requests for materials should be addressed to Y.H. (email: y-hase@mosk.tytlabs.co.jp) or to S.N. (email: nakanishi@chem.es.osaka-u.ac.jp)

The non-aqueous Li-$O_2$ battery has received much attention in the last decade due to its high specific energy, which is required for automotive applications and future energy devices[1–17]. In such units, the anodic dissolution of Li metal and the cathodic oxygen reduction reaction (ORR), accompanied by the formation of $Li_2O_2$, both occur during discharge, producing a high theoretical energy density that surpasses values obtainable from state-of-the-art lithium-ion batteries. The mechanism for the ORR-based generation of $Li_2O_2$ during discharge has been studied extensively[18–20]. During the initial step of the Li–$O_2$ battery discharge process, surface-bound $LiO_2$ is generated on the cathode surface via the one-electron ORR consisting of the reactions:

$$O_2 + e^- \rightarrow O_2^- \tag{1}$$

and

$$O_2^- + Li^+ \rightarrow LiO_2^*, \tag{2}$$

where the asterisk denotes surface-bound species. The resulting $LiO_2^*$ can partially dissolve in the electrolyte to establish an equilibrium with soluble $LiO_2^s$:

$$LiO_2^* \rightleftarrows LiO_2^s \left(\text{solvated complex} : [\text{solvent}]_n [Li^+] [O_2^-]\right). \tag{3}$$

In the case that the equilibrium lies to the left, the $LiO_2^*$ can undergo a second reduction to form insulating $Li_2O_2$ on the cathode surface (known as the surface pathway):

$$LiO_2^* + Li^+ + e^- \rightarrow Li_2O_2. \tag{4}$$

In contrast, when the solvated complex is stabilized, $LiO_2$ is precipitated via a disproportionation reaction (known as the solution pathway):

$$2LiO_2^s \rightarrow Li_2O_2 + O_2. \tag{5}$$

The discharge product, $Li_2O_2$, is insoluble in common aprotic solvents and has inherently low electron conductivity, and thus acts as an insulator on the cathode. Therefore, separation of the reaction sites at which the ORR occurs and $Li_2O_2$ is formed, which is realized by the solution pathway (Eq. (5)), is advantageous for the sustained discharge of Li–$O_2$ batteries[21–39]. That is, displacement of the equilibrium in Eq. (3) to the right by increasing the solubility of $LiO_2^*$ is necessary to improve the discharge capacity[29–40]. Previous studies have revealed several factors capable of increasing the solubility of $LiO_2^*$. As an example, the Gutmann donor number (DN) of the solvent, which is an index of Lewis basicity, is an important factor affecting the equilibrium, and it has been reported that there is a positive correlation between the discharge capacity and the DN value of the solvent[29]. It is also known that the use of additives such as $H_2O$[30], $K^+$[35], and $NH_4^+$[36] is an effective approach to increasing the solubility of $LiO_2^*$. However, the effect of these additives also depends on the solvent used, and certain solvents produce no effect. In addition, Kwabi et al. reported that the cathode potential is another important factor that determines the equilibrium[37]. Considering these results, it is evident that the main parameters that decide between the solution pathway and the surface pathway have not yet been elucidated. This is unfortunate, because a systematic understanding of the conditions determining a sustained discharge is important in the design of a Li-$O_2$ battery system with superior performance.

In the present work, we report that the discharge reaction on the positive electrode generally shows negative differential resistance (NDR) in a specific potential region of the ORR. The appearance of NDR suggests that an ORR inhibitor species adsorbs/desorbs in response to the potential. In the case of the system with large NDR, the discharge capacity is found to change dramatically as the positional relationship between NDR and the discharge potential is varied. This result suggests that it appears that the adsorption/desorption of the inhibitor is closely related to the stability of $LiO_2^*$ on the cathode. These findings provide valuable insights regarding the importance of the stability of $LiO_2^*$ in determining the discharge properties of Li–$O_2$ batteries.

## Results

**Effects of the electrolyte and the cathode on discharging.** A schematic diagram of the electrochemical cell used in the present study is shown in Fig. 1a. All components of the electrochemical cell were dehydrated prior to use, and the water content of the electrolytes, as measured by Karl Fischer titration, was confirmed to be less than 30 ppm. The cell was assembled in a glove box filled with dry argon, after which discharge tests were performed at a constant temperature of 25 °C under atmospheric conditions. Carbon paper (CP) or gold mesh (Au-mesh) was used as the substrate for the positive electrode. Other details of the experimental protocols are described in the section of Methods.

Figure 1b presents representative discharge profiles on CP obtained in electrolytes consisting of 0.5 M lithium bis(trifluoromethanesulfonyl)imide (LiTFSI) in MeCN, DMSO, N,N-diethyl-N-methyl-N-(2-methoxyethyl)ammonium bis(trifluoromethanesulfonyl)imide (DEME-TFSI) or tetraethylene glycol dimethyl ether (TEGDME) at 0.2 mA cm$^{-2}$. The results shown here are partially consistent with those provided in a previous report in which Au was used as the positive electrode[29]. The discharge capacity generally increases in order of the DN value: DEME-TFSI (green, DN = 10)[41], TEGDME (black, DN = 17)[18], and DMSO (blue, DN = 30)[18]. However, the discharge capacity obtained using MeCN/CP, which has a relatively low DN of 14[18], was much greater than that observed with DMSO/CP, which has a high DN. A typical discharge profile acquired in MeCN/CP system, exceeding 200 mAh g$_{cathode}$$^{-1}$ and representing a five-fold increase in capacity compared to DMSO/CP system, is shown in Fig. 1b (red). These data also demonstrate that the potential sometimes oscillated in MeCN/CP, and this peculiar phenomenon is discussed further on.

The discharge capacities in the DMSO/CP and MeCN/CP systems acquired during the experimental trials were in the ranges of 25–54 mAh g$_{cathode}$$^{-1}$ and 200–270 mAh g$_{cathode}$$^{-1}$, respectively. Thus, the capacities obtained in MeCN/CP system were always much larger (greater by factors of 4 to 10) than those obtained in DMSO/CP system at 0.2 mA cm$^{-2}$ (Supplementary Fig. 1). Previous reports have shown that the thickness of the $Li_2O_2$ growth via the surface pathway is limited to approximately 6 nm because of the insulating properties of the oxide[42,43]. The specific surface area of the CP cathode was 0.31 ± 0.20 m$^2$$_{BET}$ g$^{-1}$, and so the maximum capacity via the surface pathway was calculated to be 3.6 mAh g$_{cathode}$$^{-1}$. The results shown in Fig. 1b imply that at least some of the $Li_2O_2$ was formed via the solution pathway in both the MeCN/CP and DMSO/CP systems. Notably, the same trend was also observed even when the cell was discharged in a glove box filled with dry argon. Based on these results, together with other supporting data discussed further on, we concluded that any potential contamination with water did not affect the results obtained in the present work (see Supplementary Information for a discussion of the possible effects of contamination by water). It was also confirmed that the specific Li salt used was not a factor in obtaining a prolonged discharge, by changing the Li salt from LiTFSI to lithium perchlorate (LiClO$_4$) (Supplementary Fig. 2). It is noteworthy that potential oscillations do not appear in Supplementary Fig. 2, indicating that these

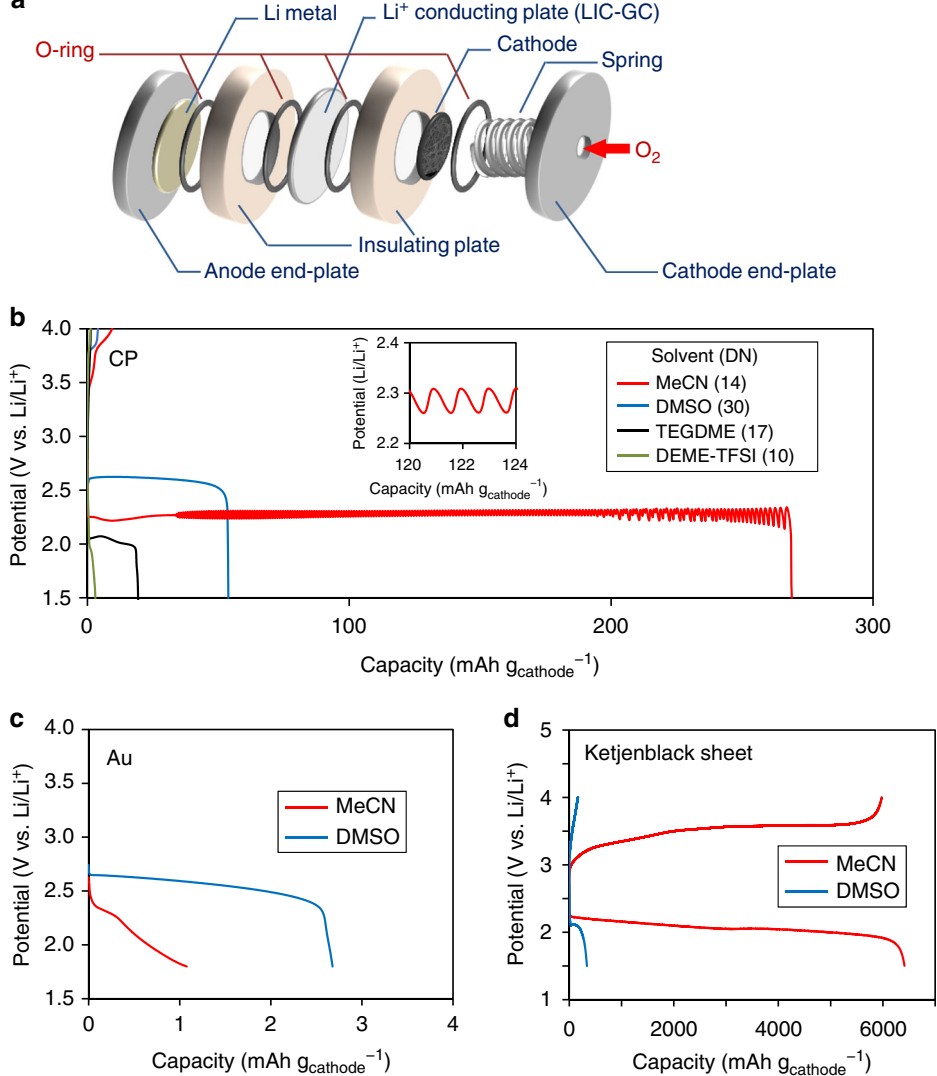

**Fig. 1** Discharge and charge curves of Li-O$_2$ cells employing various solvents with 0.5 M LiTFSI electrolytes. **a** The schematic illustration of an electrochemical cell for a Li-O$_2$ battery. **b-d** The discharge and charge curves of the Li-O$_2$ cells employing carbon paper (CP) (**b**), Au-mesh (**c**), and Ketjenblack sheet (**d**) as cathodes. The applied current densities were 0.2 mA cm$^{-2}$ (**b**), 0.02 mA cm$^{-2}$ (**c**), 1.25 mA cm$^{-2}$ (**d**, discharge), and 0.25 mA cm$^{-2}$ (**d** charge). **b** The inset shows the magnified image of **b**

oscillations were not a necessary condition for the sustained discharge in the MeCN/CP system.

On the other hand, when the same experiment was conducted with the cathode substrate changed from CP to Au-mesh, the capacity in MeCN was smaller than that in DMSO (Fig. 1c), in good agreement with the previous study[29]. (Note that the effective surface area of the Au-mesh electrode is significantly smaller than that of the CP electrode. See the section of Methods for details.) These results indicate that an appropriate combination of electrolyte solvent and cathode substrate is critical for significant improvement of the discharge capacity. This is an essential point we claim in this research, and will be discussed carefully later.

Although the discharge capacity was considerably enhanced in the MeCN/CP system at a higher current density of 0.2 mA cm$^{-2}$, the charging reaction did not proceed and the coulombic efficiency was only 3.5% (Fig. 1b). This low coulombic efficiency was due to the two-dimensional nature of the CP substrate, which was unable to re-trap all the LiO$_2^§$ formed in the electrolyte and allow this compound to precipitate on the cathode substrate. The

coulombic efficiency was confirmed to be significantly improved upon employing a Ketjenblack sheet cathode (Fig. 1d).

**Characterization of the discharge products.** Figure 2a, b presents respectively the results of Raman and powder X-ray diffraction (PXRD) analyses of the products deposited on CP during the discharge reaction in an MeCN electrolyte containing 0.5 M LiTFSI at 0.2 mA cm$^{-2}$. In both cases, it was confirmed that Li$_2$O$_2$ was formed as the discharge product. Figure 3a and Supplementary Figs. 3a and 3b present scanning electron microscopy (SEM) images of the cathode extracted from the cell of the DMSO/CP system after discharge at a capacity of 23 mAh g$_{cathode}^{-1}$. The CP surface is seen to be fully covered with Li$_2$O$_2$ flakes. In contrast, in the MeCN/CP system, Li$_2$O$_2$ was deposited non-uniformly and many parts of the CP surface remained bare, as shown in Fig. 3b and Supplementary Figs. 3c and 3d. These images were obtained using the MeCN/CP system after discharging under the same conditions as used to obtain the specimen in Fig. 3a. The sample examined in the case of a discharge capacity

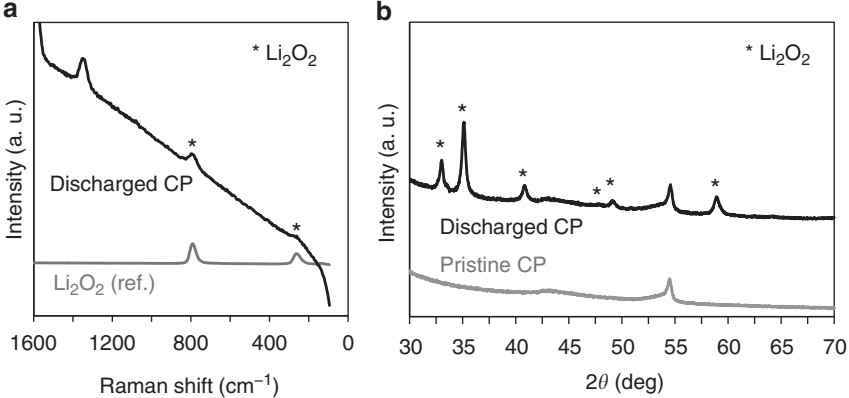

**Fig. 2** Characterization of the discharged obtained from the MeCN system. **a** Raman spectra of the discharged CP and $Li_2O_2$ reagent as a reference. **b** PXRD patterns of CP obtained before and after discharging

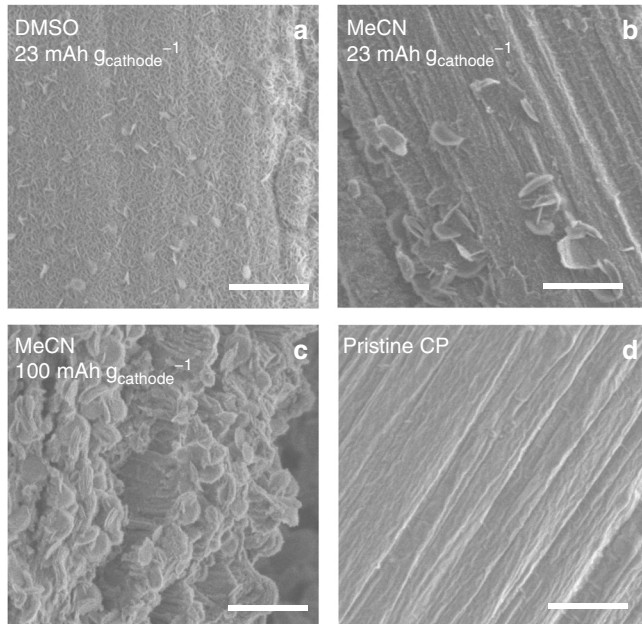

**Fig. 3** Solvent and cathode substrate effects on morphology of discharge product. SEM images of CP obtained after discharge in the DMSO (**a**), MeCN (**b**, **c**), and pristine state of CP (**d**). All scale bars, 1 μm

of 100 mAh $g_{cathode}^{-1}$ exhibits more columnar-shaped deposits (Fig. 3c), which strongly suggest that $Li_2O_2$ was preferentially formed via the solution pathway in the MeCN/CP system at 0.2 mA cm$^{-2}$.

The amounts of $Li_2O_2$ formed and $O_2$ consumed during discharge were subsequently quantified, and the associated $e^-/O_2$ ratios were estimated. It was confirmed from the $O_2$ consumption values calculated from the pressure inside the cell during discharge that the $e^-/O_2$ ratio during the discharge reaction was ~2.0 (Supplementary Fig. 4). The yields of $Li_2O_2$ ($Y_{Li2O2}$) based on the discharge capacity values were compared to values determined using the iodometric titration method[44]. The amounts of $Li_2O_2$ produced corresponded to approximately 80% of the values estimated from the discharge capacity, and the ratio of $Li_2O_2$ formation to the electrochemical reduction reaction was equivalent to that reported in a previous publication (Supplementary Table 1)[44–47]. The missing 20% of the discharge capacity that was not evident from the iodometric titration results

is attributed to various previously reported side reactions. On the basis of these quantitative experiments, it was concluded that $Li_2O_2$ formation proceeded primarily via the ORR in Eq. (1), in agreement with previous reports.

**Negative differential resistance**. As noted, a spontaneous potential oscillation appeared over a specific current density range in the case of the MeCN/CP system, in which a significantly higher discharge capacity was obtained (Fig. 1b and Supplementary Figs. 5 and 6). This potential oscillation was not a necessary condition for the sustained discharge observed in the MeCN/CP system. However, the appearance of these oscillations is important with respect to understanding the origin of the sustained discharge in the low-DN MeCN/CP system[29], as discussed further on.

NDR is typically observed in electrochemical systems that exhibit spontaneous oscillations of current and/or potential[48]. The existence of NDR in the battery systems that we investigated is evident from the potential oscillation in Fig. 1b and from the results obtained from the electrochemical analysis of the ORR, as described below. Figure. 4a shows two consecutive cyclic voltammetry (CV) cycles acquired in MeCN. It should be emphasized that the CV data in Fig. 4a were obtained at a slow scan rate of 0.5 mV s$^{-1}$, and CV data essentially equivalent to those in the literature were obtained when the scan rate was increased to 100 mV s$^{-1}$ (Supplementary Fig. 7). The absolute value of the cathodic current ($I_{abs}$) initially increased during the negative sweep and reached a peak at approximately 2.5 V (the region between points a and b in Fig. 4a). A second reduction peak then occurred at lower potentials (points b–c). During the positive sweep, a constant $I_{abs}$ was observed in the region between points c and d. However, when the voltage exceeded approximately 2.3 V, $I_{abs}$ began to slightly increase, despite the decrease in the overpotential (points d–e), which clearly indicates the presence of the NDR. The onset of NDR was also confirmed for the DMSO systems as well during trials in which the positive electrode was changed from CP to Au (Supplementary Fig. 8). These results indicate that NDR is inherent to the positive electrode reaction during the discharge process.

The second reduction peak observed in this study is typically assigned to the reductive formation of $Li_2O_2$[18–20,29]. Although $Li_2O_2$ will not decompose in the potential range over which the CV data were acquired in the present work, $I_{abs}$ increased between points d and e. In addition, the $I_{abs}$ value was almost the same as that during the previous cycle (Fig. 4a). Thus, it appears

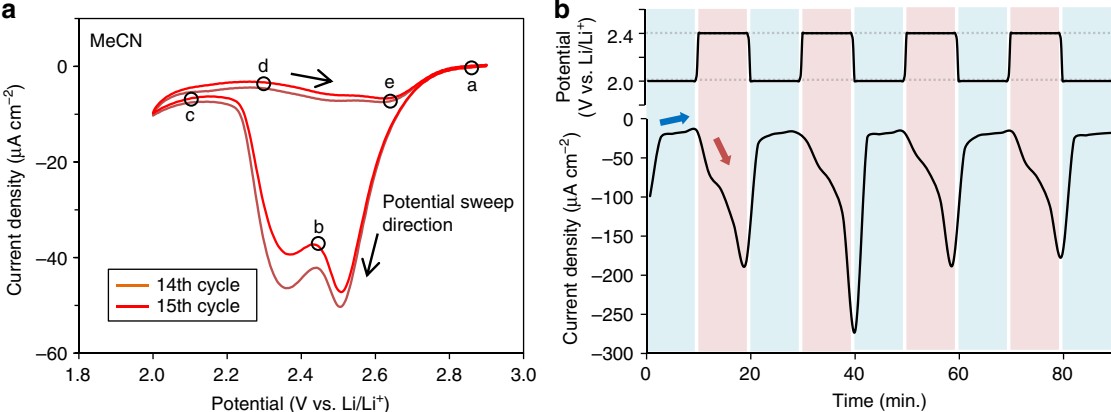

**Fig. 4** Negative differential resistance (NDR) of ORR potential region. **a** The CVs of the 14th and 15th cycles for MeCN system using 2-electrode set up with CP for working electrode. The potential scan rate was 0.5 mV s$^{-1}$. The current increase between point d and e in the positive sweep shows the NDR. **b** The time course of the ORR current for the MeCN system at the constant potential of 2.0 and 2.4 V obtained after 10 cycles of CV measurement. The potential was switched at 10-min intervals

that a reaction-inhibiting species other than $Li_2O_2$ was reversibly adsorbed and desorbed on/from the cathode surface in response to the potential change. The $2e^-$ reduction reaction to form $Li_2O_2$ was also slow in this potential region, as has been previously suggested based on in situ spectroscopic investigations[49]. Thus, these CV results confirmed that $I_{abs}$ increased (or decreased) with a decrease (or increase) in the overpotential within a specific potential region, demonstrating that NDR appears in this electrochemical system. The presence of NDR (and the reversible activation/passivation of the cathode) was further confirmed by observing the time course of the $I_{abs}$ at different potentials. Figure 4b shows that $I_{abs}$ increased and decreased repeatedly when the cathode potential was alternated between 2.0 and 2.4 V using the MeCN/CP system, providing further evidence for NDR.

**NDR potential region and correlation with discharge capacity.** As detailed above, the ORR in aprotic electrolytes containing $Li^+$ ions typically shows the occurrence of NDR. Meanwhile, it has been reported that $Li_2O_2$ is preferentially formed via either the solution or surface pathways at higher or lower potentials, respectively[37,38]. Based on the above, we hypothesized that the transition from the solution pathway to the surface pathway results in the expression of NDR. To verify this hypothesis, it was necessary to investigate the correlation between the working potential of the positive electrode and the discharge capacity. However, in general, the relationship between the set and observed potential in an experiment, $U$, and the true electrode potential (or Helmholtz double layer potential), $E$, that is a critical factor determining electrode reaction mechanisms, will differ based on the ohmic drop ($IR$, where $R$ is the resistance of the solution between the electrode surface and the reference electrode and $I$ is negative for the reduction current) according to the equation $U = E + IR$. Furthermore, during the discharge of a Li-$O_2$ battery, $Li_2O_2$, which is a poor conductor, accumulates and gradually covers the cathode surface so that the $IR$ value changes with time during the discharge process. Because of this complicated set of interconnections, potentiostatic experiments are more suitable for a qualitative evaluation of the correlation between the NDR region of the true electrode potential ($E_{NDR}$) and the discharge capacity, than potentiodynamic experiments shown in Fig. 4a and Fig. 1a. For this reason, we conducted discharge tests under constant potential conditions, as described below.

Figure 5b presents the discharge profiles acquired at different potentials employing the MeCN/CP system. At a positive electrode potential lower than 1.9 V, a large current was initially observed but gradually decreased with time. The final discharge capacity was only approximately 2.5 mAh $g_{cathode}^{-1}$ (cut-off current density: 0.05 mA cm$^{-2}$). Conversely, at a positive electrode potential higher than 2.0 V, a small but steady current continued to flow and the discharge capacity reached more than 110 mAh $g_{cathode}^{-1}$. The discharge capacities measured at various potentials using the MeCN electrolyte are summarized in Fig. 5d (circles). These discharge capacities undergo a significant change in the vicinity of 1.9 to 2.0 V. When the positive electrode potential was varied across this potential, it was confirmed that the current reversibly increased and decreased, just as in Fig. 4b (Supplementary Fig. 11), indicating the presence of NDR in this potential region. Notably, the discharge capacity at lower potentials almost coincided with the theoretical value for the reaction solely by the surface pathway (the horizontal dashed line in Fig. 5d). However, a large discharge capacity exceeding 100 mAh $g_{cathode}^{-1}$ was obtained at higher potentials, suggesting that the solution pathway was predominant in this potential region (see Supplementary Information for a more detailed discussion of the relationship between the NDR and the reaction pathway).

Qualitatively similar potential-dependency data were obtained for the DMSO/CP system, as shown in Fig. 5c and d. As previously observed in Supplementary Fig. 8, the NDR also occurs in the DMSO/CP system although, unlike the MeCN/CP results, the NDR is not clearly evident in Fig. 5d (squares). Nevertheless, the discharge capacity is evidently lowered as the potential becomes negative. Another notable difference from the MeCN/CP system is that the discharge capacity is much greater than 3.6 mAh $g_{cathode}^{-1}$ (dashed line) even at lower potentials (Fig. 5d). These results suggest that the solution pathway was dominant over the entire potential region in the DMSO/CP system, although the proportional contributions of the solution and surface pathways depend on the potential, which has NDR characteristics.

The $E_{NDR}$ value is, in principle, determined by the adsorption isotherm of the inhibitor. That is, $E_{NDR}$ will vary with the system components, such as the electrolyte and electrode material. In addition, as described earlier, $U$ and $E$ will differ due to the ohmic drop, the value of which changes with the electrode structure and with past battery operation. Therefore, the NDR potential region is not absolutely set and will appear in a specific region in

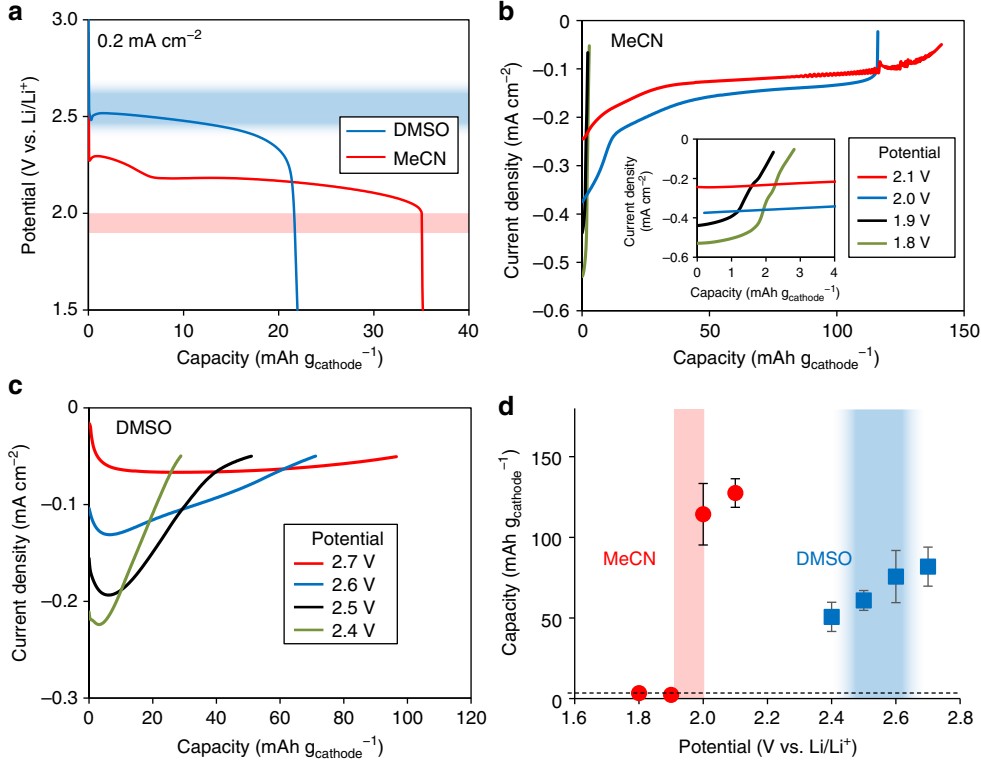

**Fig. 5** Estimation of NDR potential region. **a** Galvanostatic discharge curves obtained in 0.05 M LiTFSI electrolytes. The applied current densities were 0.2 mA cm$^{-2}$. **b**, **c** Potentilstatic discharge curves obtained in 0.05 M LiTFSI electrolytes with pre-discharging. The cut off current density was 0.05 mA cm$^{-2}$. **b** The inset shows the magnified image of **b**. **d** Discharge capacities of potentiostatic discharging at various potentials corresponding to Fig. 5b (MeCN, red circles) and 5c (DMSO, blue squares). The horizontal dashed line shows the capacity supposed to be obtained with surface pathway (3.6 mAh g$_{cathode}^{-1}$)

different systems. Based on this, we can qualitatively explain the larger discharge capacity obtained from the MeCN electrolyte (having a lower DN) compared to the DMSO electrolyte (having a high DN). Figure 5a demonstrates that the operating potential for the MeCN system is in the range of 2.2 to 2.3 V (that is, higher than the NDR potential region), at which the discharge proceeds via the solution pathway. The discharge capacity is therefore significantly increased, exceeding that obtained from the DMSO system (Fig. 5d). This occurs because the operating potential is in the potential region over which the reaction proceeds via the solution pathway in MeCN, as shown in Fig. 5d

When Au-mesh was applied to the cathode, significant differences in discharge capacity were observed between DMSO/Au and MeCN/Au systems although both systems showed a gentle potential dependence of the discharge capacity as shown in Supplementary Fig. 13. The average discharge capacities at each potential for the MeCN/Au system were around 60 μAh g$_{cathode}^{-1}$, while the higher capacity than 100 μAh g$_{cathode}^{-1}$ were obtained for the DMSO/Au system when the measurement potentials were set above 2.2 V (cut-off current density: 0.005 mA cm$^{-2}$). The theoretical maximum capacity via surface pathway on Au-mesh, which was based on the specific surface area of Au-mesh ($8.7 \times 10^{-3}$ m$^2_{BET}$ g$^{-1}$), was estimated at 61 μAh g$_{cathode}^{-1}$, therefore, the results shown in Supplementary Fig. 13 suggest that the surface and solution pathways were dominant in the MeCN/Au and DMSO/Au systems, respectively. The conclusions regarding the Au-mesh systems derived from Supplementary Fig. 13 are in good agreement with reports in the literature that Li$_2$O$_2$ on Au electrode formed mainly via the solution pathway in the high-DN DMSO electrolyte, while the

surface pathway is dominant in low-DN electrolyte, MeCN, at all potentials[29].

Here, let us briefly summarize the similarities and differences in all the systems (i.e., MeCN/CP, DMSO/CP, MeCN/Au, and DMSO/Au systems). As described above, Supplementary Fig. 8 was the clear evidence that the ORR in Li–O$_2$ batteries inherently involve the NDR. However, the magnitude of NDR and potential dependency of the discharge capacity varies with the combination of electrolyte and electrode substrate of each system which is classified in three types, Type-1, 2 and 3 for the later discussion. The MeCN/CP system, which is classified as Type-1, showed a clear potential dependency of the discharge capacity and a large NDR (Fig. 5d, circles). For the other three systems, unlike the MeCN/CP system, both of the NDR and the potential dependence of the discharge capacity were not clear as those in the MeCN/CP system. The discharge capacity in the DMSO/CP and DMSO/Au systems was significantly larger than the maximum capacity via the surface pathway, indicating that the solution pathway is dominant in these two systems (Type-2). Meanwhile, the discharge capacity for the MeCN/Au system was nearly equivalent to the theoretical capacity via surface pathway at all potentials (Type-3).

**Possible origin of NDR and correlation with the reaction pathway.** The Faraday current ($I_F$) in an electrochemical system can be expressed as the product of the number of electrons ($n$), the Faradaic constant ($F$), the reaction rate constant ($k$), the effective surface area ($A$), and the concentration of electroactive species ($c$). Because $n$, $F$, and $c$ are all constants, the differential

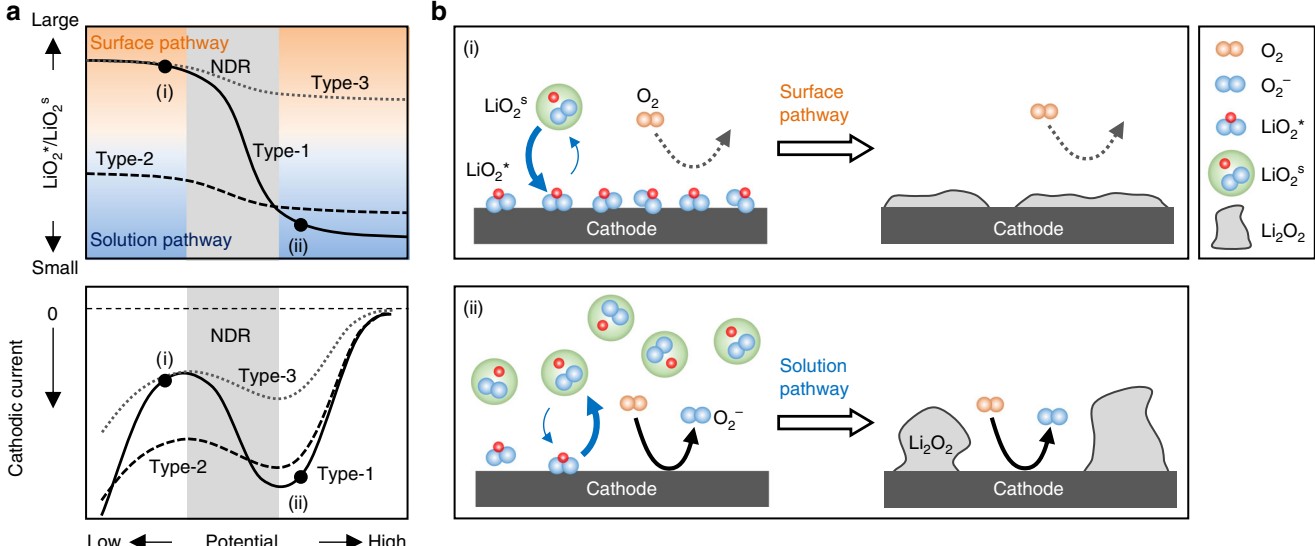

**Fig. 6** Schematic illustration of the proposed mechanism for the formation of $Li_2O_2$. **a** The changes of ratio, $LiO_2^*/LiO_2^s$ (above) and cathodic current (below) with potentials. The NDR potential region indicates the transition region of the ratio ($LiO_2^*/LiO_2^s$) in each system. **b** The numbers of illustration (i) and (ii) are corresponded with the numbers plotted in **a**. In the case of Type-1, the surface pathway is dominant at the state (i) of lower potential than the NDR potential region. In contrast, the solution pathway is encouraged at the state (ii) in higher potential region than NDR

resistance in an electrochemical system can be expressed as[48]:

$$dI_F/dE = d(nFckA)/dE = nFCk\, dA/dE + nFCA\, dk/dE. \quad (6)$$

Therefore, for NDR to appear, either $dA/dE < 0$ or $dk/dE < 0$ must be true. Each of these two conditions can be induced with an increase of the overpotential by an increase in the coverage of the reaction-inhibiting adsorbed species or by a decrease in the coverage of the reaction-promoting adsorbed species. In the present work, the current was found to decrease over time during the discharge of Li-O$_2$ batteries, indicating that the former mechanism was associated with the appearance of NDR.

At this point, it is helpful to consider the identity of the reaction-inhibiting species. It is known that the cathode potential partly affects the equilibrium in Eq. (3). More specifically, the extent of coverage by $LiO_2^*$ is larger or smaller at lower or higher cathode potentials, respectively, which determines whether the surface or solution pathway occurs[29,37]. Density functional theory (DFT) calculations have also suggested that the stability of $LiO_2^*$ is affected by potential[38]. Furthermore, it is believed that the ORR cannot proceed on $LiO_2^*$ because $O_2$ must be adsorbed on the active sites of the cathode during the ORR. Therefore, as the potential is lower (or higher), the coverage of $LiO_2^*$ becomes larger (or smaller), and eventually the ORR current becomes smaller (or larger). In other words, the system exhibits NDR characteristics. On this basis, $LiO_2^*$ is considered to be the most likely ORR inhibitor (Fig. 6a). However, it should be noted that the precise identity of the adsorbed species was not identified by spectroscopic means in this work.

## Discussion
The correlation between the $LiO_2^*/LiO_2^s$ ratio and the potential, which represents the $LiO_2^*$ adsorption isotherm and $E_{NDR}$, varies with the system components such as the electrolyte and electrode material. Figure 6a schematically shows the $LiO_2^*$ adsorption isotherm with respect to the potential in the ORR region. The adsorption isotherms are classified into the following three types. That is, Type-1 in which the ratio of $LiO_2^*/LiO_2^s$ varies largely below and above the NDR region, Type-2 in which the ratio is

small at all potentials, and Type-3 where the ratio is totally large. The results of potentiostatic discharging shown in Fig. 5 and Supplementary Fig. 13 provided a clue to how each system draws its specific adsorption isotherm. The MeCN/CP system, which is attributed to Type-1, the reaction proceeded predominantly via the surface pathway at an operating potential lower than the NDR potential region resulting in inevitable early cell death (Fig. 6b-(i)). In contrast, at an operating potential higher than the NDR potential region, the solution pathway become relatively dominant, which led to an enhanced discharge capacity (Fig. 6b-(ii)). The formed $LiO_2^s$ disproportionates immediately to form $Li_2O_2$ in the MeCN electrolyte because MeCN has low DN. However, the deposition of $Li_2O_2$ and the ORR of Eq. (1) proceed at the different positions of the cathode, therefore, the whole surface of the cathode cannot be covered by the deposition, as shown in Fig. 3b, resulting in a higher discharge capacity. Meanwhile, the DMSO/CP and DMSO/Au systems draw the adsorption isotherm of Type-2 where the solution pathway was encouraged at all potentials. The adsorption isotherm for the MeCN/Au-mesh system is assigned to Type-3 where the surface pathway was fully dominant (Fig. 6a). Differences in the geometry of the CP and Au-mesh might affect the actual adsorption isotherm. However, the result that the discharge capacity obtained in MeCN/CP system was larger than that in DMSO/CP in a specific potential region implies that the difference in adsorption isotherm derived from the cathode material is more essential.

It is noteworthy that the extent of NDR in the MeCN/CP system was significantly higher than that in the other three systems of DMSO/CP, MeCN/Au, and DMSO/Au systems, although NDR was commonly observed in all of them. Given that electrochemical oscillations tend to occur more readily in systems having more pronounced NDR[48], it is not unexpected that electrochemical oscillations were observed only in the MeCN system (Fig. 1b and Supplementary Figs. 5 and 6). A detailed mechanism for these oscillations is provided in Supplementary Information.

In conclusion, we demonstrated that ORR-inhibiting species are generated during the discharge of Li-O$_2$ batteries, resulting in the appearance of NDR. When the operating potential is higher than the NDR potential region, more continuous discharge is

obtained as a result of the solution pathway because of the reduced coverage of the cathode by the inhibitor. It is evident that the NDR potential region is an important factor determining the continuity of the ORR. Considering the behavior of the inhibitor formed along with the ORR on the cathode and adsorption/desorption in the potential region of the ORR, the inhibiting species is most likely $LiO_2^*$ or a related compound. In principle, the equilibrium potential of Eq. (3), and thus the NDR potential region, is determined by the relative stabilities of $LiO_2^*$ and $LiO_2^\S$. The stability of $LiO_2^\S$ is affected by the DN value of the solvent and also by the presence of additives or contaminants, as previously reported, while the stability of $LiO_2^*$ is determined by the electrolyte as well as the cathode material and potential. A greater discharge capacity can be achieved even in a solvent having a low DN value if the cell is operated such that the cathode potential is more positive than the NDR potential region when the system shows a large NDR. Our results demonstrated that the discharge capacity can be increased higher than the value which is obtained in a solvent with a high DN even in a solvent having a low DN such as MeCN by selecting the appropriate operating potential. The operating potential is generally determined by the balance between the applied current density and the external load, the latter of which is affected by various factors, including reaction kinetics and the geometrical structure of the cathode. Therefore, these parameters should be adjusted accordingly for sustained discharge along with verifying the specific NDR for each system. This fundamental study of the ORR in Li–O2 batteries is expected to lead to an improved understanding of the underlying chemistry, which is necessary to realize the development of more sophisticated Li–O2 units.

## Methods

**Electrochemical measurements.** Assembly of cells and introduction of oxygen gas were carried out in the glovebox filled with dry Ar gas, using fully dried instruments and materials. All electrolytes were dried for several days over freshly activated molecular sieves. The water content of the prepared electrolytes, measured by Kerl Fischer titration, are <30 ppm except for the DEME-TFSI electrolyte (the water content of DEME-TFSI electrolyte is <60 ppm). A typical experimental procedure is described below. The anode and cathode compartments of the Swagelock-type cell were separated by a solid-state Li ion-conductive glass-ceramic film (LIC-GC, Ohara Inc., 150 μm of thickness) to prevent side reactions between the Li metal and electrolytes applied to the cathode side (Fig. 1a). This cell system allows us to employ MeCN as an electrolyte solvent to a Li–O2 battery with Li metal anode, even though MeCN shows crucially poor reductive stability. CP (TGP-H-060, Toray Industries, Inc., 190 μm of thickness) or Au-mesh (Nilaco, 100 mesh) was used as a cathode electrode. After drying CP cathodes in vacuo at 150 °C for 5 h, the cathodes were transferred to a glove box without exposure of ambient air. Au-mesh was cleaned in acetone with ultrasonic cleaning apparatus before use. The average masses of CP and Au-mesh were 0.0260 g and 0.180 g, respectively. The specific surface areas were $0.31 \pm 0.20$ m$^2_{BET}$ g$^{-1}$ and 0.0087 m$^2_{BET}$ g$^{-1}$, respectively, which were measured by Kr (99.999%) physisorption with a BELSOAP-MAX (MicrotracBEL) at 77 K employing the Brunauer-Emmett-Teller isotherm. 200 μL of 0.5 M LiTFSI (Tokyo Kasei) in MeCN (Wako) was added into the cathode-side chamber of the cell and high-purity O2 gas (0.15 MPa, > 99.99%) was introduced into a 50 mL cylinder which was attached to the cell during the electrochemical measurements. Li foil (Honjo Metal, 0.4 mm of thickness) and 1 M LiTFSI in the mixed solvent of EC and DEC (Kishida Chemical) were used as an anode and an electrolyte for the anode-side, respectively. After that, the assembled cells were taken out from the glovebox, and the battery characteristics were determined using an Aska Electronic ACD-01 battery performance analyzer under atmospheric condition at 25 °C. The cell was allowed to rest at open circuit voltage for at least 12 h before measurements. The galvanostatic discharging experiments were performed to a reductive potential of 1.5 V at a current density of 0.2 mA cm$^{-2}$.

Potentiostatic discharging was performed using Li-O2 cell setup which was assembled in the same way described above. The potentiostatic discharge characteristics were determined using a Bio-Logic VMP3 system at 25 °C. For the pre-discharging experiment of the MeCN/CP system, 1 mAh g$^{-1}$ of discharging was performed at 2.4 V. After that, the potentials of the cells were immediately shifted to various potentials for the potentiostatic discharging test and potentiostatic discharging was performed until the current density reached 0.05 mA cm$^{-2}$. For the DMSO/CP system, the pre-discharging experiment was carried out for 1 mA g$^{-1}$ at 2.7 V.

CV (2-electrode setup) was performed using the Li–O2 cell setup which was similar to the above and HZ-5000 (Hokuto) at 25 °C.

**Characterization of the discharged cathodes.** For the characterization of discharged cathodes, the cells were disassembled in an Ar glovebox and the discharged cathodes were extracted from the cells. The cathodes were immersed in anhydrous MeCN for 24 h and rinsed with MeCN for three times to remove electrolytes and dried in vacuo at room temperature for 2 h. SEM images were obtained using SEM system (Hitachi high-technologies, S-4300) at an acceleration voltage of 1 kV. The samples were transferred from a glove box to the SEM chamber using a closed vessel to avoid exposure to the air. Powder X-ray diffraction (PXRD) patterns of the discharged cathodes were measured with a powder X-ray diffractometer (Rigaku, UltimaIV) in an air-sensitive sample holder. Measurements were performed at room temperature from 5 to 70° $2\theta$. Raman spectra were measured in an airtight quartz cell on a Raman spectrometer (JASCO, NRS-3200).

## Data availability

The authors declare that data supporting the findings of this study are available within the paper and its supplementary information file or from the corresponding author on reasonable request.

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

## Acknowledgements

The authors thank Dr. S. Matsuda and Dr. K. Takechi for valuable discussions. The authors are also grateful to Dr. Y. Kato and Dr. N. Setoyama for acquiring Raman spectra and Brunauer-Emmett-Teller surface area data, respectively, as well as to Mr. T. Uyama for performing PXRD measurements.

## Author contributions

All authors contributed to the design of the research and performed the experimental data analysis. Y.H., Y.K., T.K., J.S. and T.H. prepared materials and carried out the experimental work. T.S. and K.K. provided some scientific suggestion. Y.H., Y.K. and S. N. co-wrote the manuscript. All authors discussed the results and commented on the manuscript.

## Additional information

**Competing interests:** The authors declare no competing interests.

