## [Peer Review File · Nature Communications]

Reviewers' comments:

Reviewer #1 (Remarks to the Author):

I have reviewed the paper by Hase et al. concerning the correlation between the observation of N-type negative differential resistance and discharge capacities obtained in Li-O₂ batteries. The research presented in this paper is of topical research interest and brings in a fresh perspective that has not been discussed in the field of Li-O₂ batteries and this may be suitable for publication in Nature Communications. However, I have the following technical concerns which preclude the paper from publication, at least in its current form:

1) A major concern in this field is the cleanliness of experiments, especially in terms of water contamination. The authors did purify the solvents and measured KF titrated water contents in the solvents are less than 30 ppm. However, the CVs, especially the CVs performed with Acetonitrile do not seem to agree with previously published data from nearly anhydrous solvents. For example, the authors report a double peak in the reductive CV scan for both DMSO and Acetonitrile. However, based on past reports [See Johnson et al Nature Chemistry 6 1091 (2014) and Aetukuri et al Nature Chemistry 7 50 (2015)], in nearly anhydrous Acetonitrile, a single reductive peak has been reported. Similar data was obtained for DME as well. Double peaks in reduction have been observed either in DMSO, where solution-mediated deposition was predominant, or in solvents that contained water. The authors should definitely discuss the origin of these differences.

2) If the presence of the second reductive peak (at a lower potential) in Acetonitrile is due to solvent contamination with water, do the conclusions still hold?

3) In the presence of water, LiOH has also been shown to form in aprotic Li-O₂ batteries [(Liu et al Science 350 530 (2015))]. In this scenario, Would LiO₂* still be the passivating layer that gives rise to NDR. Why can't one posit that the passivation is due to LiOH?

4) There appears to be a shoulder in the XRD peak at ~32 degrees in the XRD pattern of the discharge product. What is the origin of this? LiOH?

5) The authors observed cathode dependent changes in the ultimate capacity. The origins of this are unclear. Why would a change in the cathode surface change the growth mechanism from being surface growth to solution growth? In my opinion, understanding this is essential to the conclusions of the paper.

6) A minor point: In page 3 line 18, the chemical name should read tetraethylene

Reviewer #2 (Remarks to the Author):

The paper describes a study of the effect of electrolyte and electrode material on the capacity of the Li-O₂ battery. It is proposed that with a combination of carbon paper, acetonitrile, and discharge at a specific potential in the NDR region a high capacity, far in excess of that obtained from DMSO is obtained. Broadly I agree with the main observation that acetonitrile can provide a very high capacity in some instances, but perhaps not the reason proposed here. The NDR region is proposed to occur due to adsorption of LiO₂ at negative potentials and discharge at a voltage within or negative of the NDR region is proposed to enable discharge by a solution mechanism. However, it is not clear from

the discussion what exactly is happening here. The data doesn't prove any real relationship between the NDR region, oscillation and the improved performance with acetonitrile. In addition, while stability of adsorbed LiO₂ will increase towards negative potentials (higher overpotentials for this reaction), the same can be said for Li₂O₂. Finally, there are some problems with the interpretation of the electrochemistry which limit confidence in the proposed theory. In summary I do not consider this paper suitable for publication in Nature communications.

Major points below:

1. The need to avoid surface passivation during discharge and to select a potential where rapid passivation occurs is known. It is not clear why acetonitrile is different and that is not clear from the discussion.
2. Please clearly state the effect of the NDR and oscillation is any. If the oscillation is not responsible for the effect, please limit its discussion to enhance clarity.
3. Water in the cell could be a reason for the enhanced performance of acetonitrile. The authors should check the water content of the electrolyte after assembly of the cell.
4. In the section "mechanism of the potential Oscillation" the authors clearly describe why the voltage polarises negative. This is already known. They state "The rapid change of the potential from the negative- to positive-end of the potential oscillation can be explained in the exactly same way.". Please also describe in detail the reason for the shift to positive potentials.
5. For all cells, a chemical percentage yield measurement (Li₂O₂ measured / Li₂O₂ expected from capacity). A number of these can be found in the literature. This will confirm that the acetonitrile cell is forming Li₂O₂ and not a side reaction. A measure of the O₂/e ratio would also help, i.e., using a pressure cell.
6. It is interesting, and strange the acetonitrile cell appears to form larger Li₂O₂ structures by a solution mechanism. Acetonitrile is not expected to do this as LiO₂ has a very poor solubility in this liquid. The authors should very carefully check water and decomposition reactions carefully.
7. The authors should give consideration to the effects of viscosity, which may enable a solution reaction acetonitrile even with low solubility, assuming the current is low enough.
8. O₂ reduction in Li acetonitrile does not typically show two peaks and the voltammograms has an appearance of being wet. Further analysis is needed here, i.e. is the first peak quasi-reversible?
9. Voltammetry on a high surface area porous electrode in a cell for cycling is not ideal for electroanalysis. For example, how can the authors confirm that a peak due to depletion of O₂ in the porous structure has not resulted in a peak, followed by further result due to standard linear diffusion to the electrode surface. Moreover, it is suggested that the current does not drop during cycling, even though oxidation is not performance. However, as a high surface area material is used, it is not clear that all the area will be consumed on a cycle. i.e., total surface area may not be limiting the current. This may explain the increase current between d and e on the positive scan. In summary, these data need repeating in a 3-electrode configuration in a standard electrochemical cell at a planer electrode, before these conclusions can be confirmed.
10. Scan rates and salt concentrations should be added to the legend. Plots should use current density. mA cm⁻².

Reviewer #3 (Remarks to the Author):

This paper reports the use of negative differential resistance as an indirect indicator of the ability for Li-O₂ cathodes to accommodate large discharge capacities. Presumably, negative differential resistance (NDR) is caused by the presence of adsorbed LiO₂, which triggers the solution-phase pathway for Li₂O₂, and thus, larger discharge capacities than can be obtained by the surface pathway. While this hypothesis is intriguing and somewhat supported by the evidence, there are a number of ambiguous data points that are insufficiently explained by the authors.

1. The authors claim that higher capacities would be expected when discharge occurs within the NDR region, however the actual data from Figure 4 is somewhat ambiguous on this point. On carbon paper, in both MeCN and DMSO, it appears that all discharge potentials (with the arguable exception of 0.05 mA/cm² in MeCN) occur within the NDR region. It is thus unclear why discharge capacity in MeCN is 3-10x higher than in DMSO. It is also worth noting that the average discharge potential in the DMSO/CP system is higher (2.6-2.8 V) than in MeCN/CP (2.3 - 2.5 V). From several literature reports, it is clear that LiO₂ is more stable at higher potentials, whereas at lower potentials, it would be expected to be easily reduced at the electrode to form Li₂O₂. This expectation is opposite to what the authors find, and miniscule differences in discharge potential relative to the NDR region appear insufficient to account for this.

2. The case of Au is particularly intriguing, and reinforces the point from #1. Although discharge at 0.02 mA/cm² in the DMSO/Au system results in a higher discharge capacity, its potential is not in the NDR region (as defined by the authors), whereas the discharge potential for MeCN/Au is in the NDR region, but has a lower capacity. The authors acknowledge this inconsistency on lines 6-7 of p. 4, but provide no compelling explanation/hypothesis for the influence of NDR v potential on discharge capacity.

3. Have the authors tried potentiostatically discharging these Li-O₂ cells at potentials chosen to be squarely within/outside the NDR to more unambiguously show its effect on discharge capacity? Such experiments would shed much light on the importance of NDR for Li-O₂ discharge capacity.

4. The analysis on p. 6 seems to attribute differences in measured current as a function of voltage to differences in the Faradaic current over time, but completely neglects any capacitive effects, which may be important where such low currents (compared to the Faradaic peaks) are recorded. How do the authors know that differences in capacitance (e.g. caused by the potential approaching the potential of zero charge) are not responsible for the reductions in current?

RESPONSE TO REVIEWER 1:

We wish to express our appreciation to the Reviewer for his or her insightful comments, which have helped us significantly improve the paper.

Comment 1: *A major concern in this field is the cleanliness of experiments, especially in terms of water contamination. The authors did purify the solvents and measured KF titrated water contents in the solvents are less than 30 ppm. However, the CVs, especially the CVs performed with Acetonitrile do not seem to agree with previously published data from nearly anhydrous solvents. For example, the authors report a double peak in the reductive CV scan for both DMSO and Acetonitrile. However, based on past reports [See Johnson et al Nature Chemistry 6 1091 (2014) and Aetukuri et al Nature Chemistry 7 50 (2015)], in nearly anhydrous Acetonitrile, a single reductive peak has been reported. Similar data was obtained for DME as well. Double peaks in reduction have been observed either in DMSO, where solution-mediated deposition was predominant, or in solvents that contained water. The authors should definitely discuss the origin of these differences.*

Response: We greatly appreciate the Reviewer's comment on this point. In accordance with this feedback, we acquired new CV data while paying careful attention to exclude water as a contaminant, and confirmed that the second reduction peak was not caused by water in the electrolyte. As the Reviewer noted, water contamination is a critical factor determining the specific pathway by which Li_2O_2 is generated in Li-O₂ batteries, and the origin of the second reduction peak has been widely discussed. In this manuscript, we focus on the occurrence of negative differential resistance (NDR) in the oxygen reduction reaction (ORR) potential region and its effect on the discharge reaction mechanism. On this basis, we performed CV measurements to investigate the NDR in detail. Typically, the adsorption/desorption reactions of inhibitor species, which are dependent on the potential, are clearly evident at very slow scan rates. In contrast, this adsorption/desorption phenomenon cannot be identified when the scan rate is sufficiently rapid relative to the responsiveness of the inhibitor. We recognize the discrepancy between the results presented in our original manuscript and previous reports, as the Reviewer pointed out. For this reason, we performed additional CV measurements under the same conditions as employed in previous studies, using a standard 3-electrode system with a glassy carbon working electrode, to clarify this critical point. Based on our results, we have added Fig. S7 and experimental details of the CV measurements to the Electrochemical Measurements section as well as to the revised Supplementary Information. The details of these supplementary CV measurements are as follows. We initially acquired CV data at a rapid scan rate. As shown in Figs. S7a to d, single and double reduction peaks were observed in acetonitrile and DMSO, respectively, at a scan rate of 100 mV/s. These results are in agreement with a prior report [Johnson et al., *Nature Chemistry* 6, 1091 (2014)], in which the second reduction peak

(at a lower potential) obtained in DMSO was attributed to the formation of Li_2O_2 in the electrolyte via the solution pathway, with a single reduction peak appearing in CV plots obtained in acetonitrile. Moreover, we examined CV data acquired using acetonitrile in the ORR potential region that corresponds to the linear sweep voltammetry measurements reported by Aetukuri *et al.* [*Nature Chemistry* 7, 50 (2015)]. The data presented in Figs. S7e and f were acquired at scan rates of 10 and 1 mV/s, respectively. Fig. S7e shows a single clear reduction peak as a result of the rapid scan rate, while double peaks were obtained at the slower scan rate (Fig. S7f). This occurred even though the CV data for the second cycle at the slower scan rate contain a single sharp peak in agreement with the results reported by Aetukuri *et al.* From these additional results, it can be concluded that the second reduction peak at 2.2 V obtained at the slower scan rate did not originate from the presence of water but rather from the adsorption/desorption of a reaction inhibitor on or from the surface of the electrode. Fig. 4a in the original manuscript provides data from trials performed with a carbon paper working electrode at the slower scan rate of 0.5 mV/s and exhibits a clear second reduction peak attributed to the onset of NDR. However, regrettably, the scan rate used to acquire the data in Fig. 4a was not provided in the original manuscript and therefore these results might be misleading. We have added the necessary details regarding the experimental conditions (as in the following text) when referencing Fig. 4a in the revised manuscript, on p. 5, line 20.

“It should be emphasized that the CV data in Fig. 4a were obtained at a slow scan rate of 0.5 mV/s, and CV data essentially equivalent to those in the literature were obtained when the scan rate was increased to 100 mV/s (Fig. S7).”

We have also added the following text and a section entitled “the Views on Possible Effects of Contamination with Water” to the revised manuscript (p. 4, lines 6-9) and to the Supplementary Information, respectively. In this text and the section, we elaborate on this crucial point.

“Based on these results, together with other supporting data discussed further on, we concluded that any potential contamination with water did not affect the results obtained in the present work (see Supplementary Information for a discussion of the possible effects of contamination by water).”

Comment 2: *If the presence of the second reductive peak (at a lower potential) in Acetonitrile is due to solvent contamination with water, do the conclusions still hold?*

Response: As noted in the prior response, we confirmed that the second reduction peak generated during CV trials in acetonitrile shown in Fig. 4a is not attributable to contamination with water but

instead results from the adsorption/desorption behavior of an ORR inhibitor species in response to changes in the potential, and therefore the conclusion in the original manuscript still holds. Regrettably, the Conclusions section of the original manuscript implied that LiO_2^* was the sole inhibiting species, which was not our intention. We have therefore revised the Conclusions section and replaced it with the following text.

“We demonstrated that ORR-inhibiting species are generated during the discharge of Li-O₂ batteries, resulting in the appearance of NDR. When the operating potential is higher than the NDR potential region (that is, U_{NDR}), more continuous discharge is obtained as a result of the solution pathway because of the reduced coverage of the cathode by the inhibitor. It is evident that the U_{NDR} value is an important factor determining the continuity of the ORR. Considering the behavior of the inhibitor formed along with the ORR on the cathode and adsorption/desorption in the potential region of the ORR, the inhibiting species is most likely LiO_2^* or a related compound. In principle, the equilibrium potential of Eq. (3), and thus the U_{NDR} value, is determined by the relative stabilities of LiO_2^* and LiO_2^{s} . The stability of LiO_2^{s} is affected by the DN value of the solvent and also by the presence of additives or contaminants, as previously reported, while the stability of LiO_2^* is determined by the electrolyte as well as the cathode material and potential. A greater discharge capacity can be achieved even in a solvent having a low DN value if the cell is operated such that the cathode potential (U) is within a specific range (i.e., $U > U_{\text{NDR}}$). The operating potential is generally determined by the balance between the applied current density and the external load, the latter of which is affected by various factors, including reaction kinetics and the geometrical structure of the cathode. Therefore, these parameters should be adjusted accordingly for sustained discharge. This fundamental study of the ORR in Li-O₂ batteries is expected to lead to an improved understanding of the underlying chemistry, which is necessary to realize the development of more sophisticated Li-O₂ units.”

Comment 3: *In the presence of water, LiOH has also been shown to form in aprotic Li-O₂ batteries [(Liu et al Science 350 530 (2015)]. In this scenario, Would LiO_2^* still be the passivating layer that gives rise to NDR. Why can't one posit that the passivation is due to LiOH?*

Response: As the Reviewer noted, LiOH is a potential inhibitor, the adsorption/desorption behavior of which depends on the potential. As pointed out, Liu *et al.* reported that LiOH was formed as a discharge product in the presence of LiI even in a non-aqueous electrolyte [*Science* 350, 530 (2015)]. However, in the present work, all experimental trials were performed in conjunction with the careful exclusion of water or other redox species such as LiI. Therefore, it is unlikely that LiOH acted as an

inhibitor during the overall discharge process, although the small amount of water that may have been present (less than 30 ppm) could have generated some LiOH. To clarify this aspect of our work, we have added an additional section entitled “Views on the Possible Effects of Contamination with Water” and the following text to the revised Supplementary Information and manuscript (p. 3, line 13), respectively.

“The cell was assembled in a glove box filled with a dry argon gas, and then subjected to discharge tests at a constant temperature of 25 °C under atmospheric conditions and carbon paper (CP) was used as the substrate for the positive electrode, unless otherwise specified.”

The main focus of this work was to demonstrate that NDR is an indicator of adsorption/desorption of the inhibitor and to introduce this concept for the first time to the field of Li-O₂ battery research. The inhibitor in turn affects the adsorption isotherm of species adsorbed on the cathode, although we did not intend to imply that this inhibitor was definitely LiO₂*. LiO₂* is the most likely candidate, but a definitive identification of the inhibitor is beyond the scope of this manuscript. To make this clear to the reader, we have partly revised the section entitled “Possible origin of NDR and correlation with the reaction pathway,” as follows (p. 8, line 22).

“At this point, it is helpful to consider the identity of the reaction-inhibiting species. The cathode potential partly affects the equilibrium in Eq. (3), meaning that LiO₂* is either stabilized or destabilized according to the cathode potential, which determines whether the surface or solution pathway occurs³⁷⁻³⁹. As discussed, the potential region in which NDR and oscillations appear is consistent with the region over which the reaction pathway switches from the solution to the surface pathway (Fig. 5). In addition, density functional theory (DFT) calculations have suggested that the stability of LiO₂* is affected by potential³⁸. These findings suggest that LiO₂* is a likely the reaction inhibitor that promotes NDR (Fig. 6a).”

Comment 4: *There appears to be a shoulder in the XRD peak at ~32 degrees in the XRD pattern of the discharge product. What is the origin of this? LiOH?*

Response: As the Reviewer noted, the XRD pattern in Fig. 2b exhibits a shoulder at approximately 32°, which is assigned to LiOH. However, we believe that this LiOH was not a discharge product but instead was generated during the sample preparation required for XRD analyses. We make this conclusion based on the following considerations. Firstly, as noted, the water concentration was maintained at or below 30 ppm during all experimental work. Moreover, as we described in the

response to Comment 1, we confirmed that the reduction peak at 2.3 V, which has been reported to originate from water contamination in an electrolyte having a low DN [Aetukuri *et al.*, *Nature Chemistry* 7, 50 (2015)], did not appear in the CV plots obtained in the acetonitrile electrolyte under conditions equivalent to those used in the previous work. We have added a section entitled “Views on the Possible Effects of Contamination with Water” to the revised Supplementary Information to communicate this point. Based on these results, we believe that a small amount of LiOH was formed during the XRD analysis process. We performed XRD analyses using carefully sealed samples, as described in the original Supplementary Information. However, the amount of LiOH in the samples evidently increased over time due to the reaction of Li₂O₂ with atmospheric water, leading to the appearance of the shoulder at 32° in Fig. 2b. We have included XRD spectra with this document that demonstrate that the LiOH (0 0 1) peak at 20.5° increased with time.

Comment 5: *The authors observed cathode dependent changes in the ultimate capacity. The origins of this are unclear. Why would a change in the cathode surface change the growth mechanism from being surface growth to solution growth? In my opinion, understanding this is essential to the conclusions of the paper.*

Response: Our work determined, for the first time, that the stability of the adsorbed species, which works to inhibit the ORR, reversibly changes depending on the potential, resulting in the appearance of the NDR. The adsorption isotherm of the adsorbed species changes both in response to the solvent properties and also the environment around the adsorbed species, such as the substrate and the structure of the electrode, as well as the temperature and potential. The stability of the adsorbed species changes not only with the solvent but with the state of the substrate surface, and so the discharge capacity varies with the cathode substrate. As a result, when switching between carbon paper and Au mesh substrates, varying discharge behavior may be exhibited in accordance with the difference in the stability of the adsorbed species on each cathode surface.

As the Reviewer noted, we used both carbon paper and Au mesh as cathode substrates and compared the resulting NDR potential regions (U_{NDR}) and discharge properties in the original manuscript. However, on further reflection, we feel that it is not appropriate to discuss the relationship between the NDR and discharge capacity based on these CV data. Since the surface state and area of the electrode were significantly different depending on whether the carbon paper or Au mesh was employed, it is difficult to accurately estimate the U_{NDR} value and to evaluate the importance of this value from the results in Figs. 4c to 4f of the original manuscript. Therefore, we have revised our conclusion to state only that NDR tends to appear in the Li-O₂ battery system. We have deleted Figs. 1d and 3d, and transferred the results obtained with the Au mesh electrode from

the original manuscript (Figs. 4e and 4f) to the revised Supplementary Information (Fig. S8) and added the following text to the revised manuscript (p. 5, line 22).

“The onset of NDR was also confirmed for the DMSO system as well as during trials in which the positive electrode was changed from CP to Au (Fig. S8). These results indicate that NDR is inherent to the positive electrode reaction during the discharge process.”

Moreover, to obtain a more accurate estimation of U_{NDR} , we performed potentiostatic discharge experiments using a carbon paper cathode. We have added the relevant figures (Figs. 5 and S12) and an associated description to the new section entitled “NDR potential region and correlation with sustained discharge” in the revised manuscript. The results of these additional trials clearly show that each system has the specific U_{NDR} and the discharge reaction pathway switches according to the positional relationship between U_{NDR} and the discharge potential.

Comment 6: *A minor point: In page 3 line 18, the chemical name should read tetraethylene*

Response: We thank the Reviewer for noting this mistake. We have corrected the error.

Thank you again for your comments on our paper. I trust that the revised manuscript is now suitable for publication.

Figure | Time course of *ex-situ* PXRD patterns. PXRD patterns of the CP cathode were obtained before (a), after discharging in the MeCN electrolyte (same spectrum of Fig. 2b) (b), and after 29 min (c), 44 min (d), and 58 min (e) of the measurement of sample (b). The diffraction patterns were indexed with references from the standard powder diffraction files of 01-085-0777 and 00-009-0355 for LiOH and Li₂O₂, respectively. All the curves are normalized to the CP's (002) XRD peak. These results indicate that Li₂O₂ formed on the cathode was reactive with water and the content of LiOH in the sample increased with time.

RESPONSE TO REVIEWER 2:

We wish to express our appreciation to the Reviewer for his or her insightful comments, which have helped us significantly improve the paper.

Comment 1: *The need to avoid surface passivation during discharge and to select a potential where rapid passivation occurs is known. It is not clear why acetonitrile is different and that is not clear from the discussion.*

Response: We appreciate the Reviewer's comment on this point. We would like to mention here that it was not our intention to suggest that acetonitrile has a different effect on discharge compared to other solvents, as noted in the original manuscript (p. 4, lines 4-8). As we noted in the original manuscript (p. 2, line 30, p.6, line 4, and p.8, line 16), it was previously reported that the Li_2O_2 formation reaction pathway depends on the cathode potential, and that the surface pathway preferentially proceeds at lower potentials [Kwabi *et al.*, *Phys. Chem. Chem. Phys.* 18, 24944 (2016)]. In this work, we determined that negative differential resistance (NDR) is an indicator of the potential region over which the reaction pathway switches, and that the NDR potential region varies depending on the cathode substrate and solvent. The intent of this manuscript was to report that NDR is an indicator of adsorption/desorption of the inhibitor and/or LiO_2^* , and to introduce this novel concept to the field of Li-O₂ battery research. We regret that we did not suitably explain the interaction between the electrode surface and the absorbed species, such that the scope of this manuscript was not clearly expressed. In the revised manuscript, we focus on the core discovery, the existence of NDR, as well as the relationship between NDR and the stability of the inhibitor and/or LiO_2^* on the cathode electrode. To make this clear, we have revised the last paragraph of the Introduction and included the following text in the revised manuscript (p. 3, lines 2 – p. 3, line 9).

“In the present work, we found that the discharge reaction on the positive electrode generally shows negative differential resistance (NDR) in a specific potential region of the ORR. The appearance of NDR suggests that an ORR inhibitor species adsorbs/desorbs in response to the potential. Because the discharge capacity was found to change dramatically as the positional relationship between NDR and the discharge potential was varied, it appears that the adsorption/desorption of the inhibitor is closely related to the stability of LiO_2^* on the cathode. These results provide valuable insights regarding the importance of the stability of LiO_2^* in determining the discharge properties of Li-O₂ batteries.”

Comment 2: *Please clearly state the effect of the NDR and oscillation is any. If the oscillation is not responsible for the effect, please limit its discussion to enhance clarity.*

Response: As we noted in the original manuscript, NDR generally occurs within the ORR potential region in Li-O₂ batteries and indicates the potential required to switch the reaction pathway. This discussion is definitely not meant to suggest that a high discharge capacity can be induced by either NDR or the oscillation phenomenon. NDR is an indicator of adsorption/desorption of the inhibitor and/or LiO₂* as we stated in the response to the Comment 1. We discussed this point in the original manuscript (sections of “Negative differential resistance (NDR)” and “Origin of the NDR and correlation with the sustained discharge”). However, because the former explanation might have been insufficient, we have made an overall revision of the both sections and replaced with the sections entitled “Negative differential resistance” and “Possible origin of NDR and correlation with the reaction pathway.” We have also added the following text to the revised manuscript (p. 8, lines 4-8):

“It is noteworthy that the extent of NDR in the MeCN system was significantly higher than that in the DMSO system, although NDR was typically observed in both. Given that electrochemical oscillations tend to occur more readily in systems having more pronounced NDR⁴⁸, it is not unexpected that electrochemical oscillations were observed only in the MeCN system (Figs. 1b, S5, and S6). A detailed mechanism for these oscillations is provided in Supporting Information.”

Comment 3: *Water in the cell could be a reason for the enhanced performance of acetonitrile. The authors should check the water content of the electrolyte after assembly of the cell.*

Response: We obtained all the experimental data while being careful to exclude contamination with water; however, we did not describe these steps sufficiently in the original manuscript. To make this evident to the reader, we have added the following text to the revised manuscript (p. 3, lines 14-18).

“All components of the electrochemical cell were dehydrated prior to use, and the water content of the electrolytes, as measured by Karl Fischer titration, was confirmed to be less than 30 ppm. The cell was assembled in a glove box filled with dry argon, after which discharge tests were performed at a constant temperature of 25 °C under atmospheric conditions.”

As the Reviewer noted, assessing the water content of the electrolyte after assembly of the cell would be valuable in terms of confirming the effect of water on our experiments. Regrettably, the

small amount of electrolyte used in our experiments (200 μL per cell, as noted in the original Supplementary Information) made it difficult to accurately determine low water concentrations (less than 30 ppm) by KF titration. As an alternative, we investigated potentiostatic discharging at various potentials. Although these trials were carried out under the same conditions as were used to obtain the data in Fig. 1, discharge capacities were obtained in conjunction with lower measurement errors at certain potentials. Importantly, in the potential region lower than that associated with the NDR, a discharge capacity equivalent to the theoretical capacity for the surface pathway (3.6 $\text{mAh/g}_{\text{cathode}}$) was obtained in the acetonitrile electrolyte. Therefore, it is evident that the effect of water on discharge properties was negligibly small. To clarify this point, we have added a section entitled “Views on the Possible Effects of Contamination with Water” to the revised Supplementary Information and inserted Fig. 5 and the following text into the revised manuscript (p. 4, lines 6-9).

“Based on these results, together with other supporting data discussed further on, we concluded that any potential contamination with water did not affect the results obtained in the present work (see Supplementary Information for a discussion of the possible effects of contamination by water).”

Comment 4: *In the section “mechanism of the potential Oscillation” the authors clearly describe why the voltage polarises negative. This is already known. They state “The rapid change of the potential from the negative- to positive-end of the potential oscillation can be explained in the exactly same way.”. Please also describe in detail the reason for the shift to positive potentials.*

Response: In accordance with the Reviewer's suggestion, we have added a detailed mechanism for the rapid change of the potential from the negative- to positive-end of the potential oscillations in the Supporting Information.

Comment 5: *For all cells, a chemical percentage yield measurement (Li_2O_2 measured / Li_2O_2 expected from capacity). A number of these can be found in the literature. This will confirm that the acetonitrile cell is forming Li_2O_2 and not a side reaction. A measure of the O_2/e ratio would also help, i.e., using a pressure cell.*

Response: Additional work was performed as suggested by the Reviewer to quantify the Li_2O_2 and O_2 consumption during discharge in acetonitrile and DMSO electrolytes. We have added these data to the revised Supplementary Information as Table S1 and Fig. S4, along with a discussion of the experimental work. The results in Table S1 show that Li_2O_2 was formed in the same yield as in

previous reports in both of the electrolytes. As in earlier studies, it was evident that the various side reactions cannot be ignored, although the e^-/O_2 ratio of approximately 2.0 was comparable to values in the literature (Fig. S4). We have therefore added the following text to the revised manuscript to explain that the acetonitrile cell formed Li_2O_2 rather than undergoing a specific side reaction (p.4, line 33 – p.5, line 8).

“The amounts of Li_2O_2 formed and O_2 consumed during discharge were subsequently quantified, and the associated e^-/O_2 ratios were estimated. It was confirmed from the O_2 consumption values calculated from the pressure inside the cell during discharge that the e^-/O_2 ratio during the discharge reaction was approximately 2.0 (Fig. S4). The yields of Li_2O_2 ($Y_{Li_2O_2}$) based on the discharge capacity values were compared to values determined using the iodometric titration method⁴⁴. The amounts of Li_2O_2 produced corresponded to approximately 80% of the values estimated from the discharge capacity, and the ratio of Li_2O_2 formation to the electrochemical reduction reaction was equivalent to that reported in a previous publication (Table S1)⁴⁴⁻⁴⁷. The missing 20% of the discharge capacity that was not evident from the iodometric titration results is attributed to various previously reported side reactions. On the basis of these quantitative experiments, it was concluded that Li_2O_2 formation proceeded primarily via the ORR reaction in Eq. (1), in agreement with previous reports.”

We have also added the following references describing the quantification of Li_2O_2 to the revised manuscript.

(44) McCloskey, B.D. et al. Combining Accurate O_2 and Li_2O_2 Assays to Separate Discharge and Charge Stability Limitations in Nonaqueous Li- O_2 Batteries. *J. Phys. Chem. Lett.* **4**, 2989-2993 (2013).

(45) Xie, J. et al. Achieving Low Overpotential Li- O_2 Battery Operations by Li_2O_2 Decomposition through One-Electron Processes. *Nano Lett.* **15**, 8371-8376 (2015).

(46) Zhang, X. et al. LiO_2 : Cryosynthesis and Chemical/Electrochemical Reactivities. *J. Phys. Chem. Lett.* **8**, 2334-2338 (2017).

(47) Qiao, Y. et al. From O_2^- to HO_2^- : Reducing By-Products and Overpotential in Li- O_2 Batteries by Water Addition. *Angew. Chem., Int. Ed.* **56**, 4960-4964 (2017).

Comment 6: *It is interesting, and strange the acetonitrile cell appears to form larger Li_2O_2 structures by a solution mechanism. Acetonitrile is not expected to do this as Li_2O_2 has a very poor solubility in this liquid. The authors should very carefully check water and decomposition reactions*

carefully.

Response: As the Reviewer noted, LiO_2 exhibits poor solubility in this electrolyte. We agree and noted this fact in the original manuscript (p. 2, lines 24-28). However, we believe that the results described below demonstrate that the large Li_2O_2 precipitates obtained in acetonitrile did not result from the presence of water or from side reactions. As we discussed in the response to Comment 3, a greater discharge capacity was obtained in the acetonitrile electrolyte even when taking special care to exclude contamination with water. As described in the response to Comment 5, the results of the additional work (in Table S1 and Fig. S4) provide quantitative values for Li_2O_2 in the acetonitrile and DMSO electrolytes. These data confirm that the occurrence of some side reactions cannot be ruled out, although the e^-/O_2 ratio was comparable to previously reported values. Moreover, acetonitrile is known to be resistant to O_2^- radicals that typically induce the decomposition of electrolytes [Bryantsev *et al. J. Phys. Chem. A* 115, 12399 (2011), Takechi *et al. ECS Electrochem. Lett.* 1, A27 (2012)]. Therefore, it is unlikely that the acetonitrile electrolyte degrades.

In any case, the aim of the present work was to demonstrate that NDR is an indicator of adsorption/desorption of the ORR inhibitor and/or LiO_2^* . This goal would not be revised even if contamination with water and/or degradation of cell components were found to affect the results. Our novel discovery in this research was that the stability of the inhibitor on the electrodes can be reversibly changed depending on the potential, resulting in the appearance of NDR. To make this clear, we have fully revised the Conclusions section and replaced it with the following text.

“We demonstrated that ORR-inhibiting species are generated during the discharge of Li-O₂ batteries, resulting in the appearance of NDR. When the operating potential is higher than the NDR potential region (that is, U_{NDR}), more continuous discharge is obtained as a result of the solution pathway because of the reduced coverage of the cathode by the inhibitor. It is evident that the U_{NDR} value is an important factor determining the continuity of the ORR. Considering the behavior of the inhibitor formed along with the ORR on the cathode and adsorption/desorption in the potential region of the ORR, the inhibiting species is most likely LiO_2^* or a related compound. In principle, the equilibrium potential of Eq. (3), and thus the U_{NDR} value, is determined by the relative stabilities of LiO_2^* and LiO_2^s . The stability of LiO_2^s is affected by the DN value of the solvent and also by the presence of additives or contaminants, as previously reported, while the stability of LiO_2^* is determined by the electrolyte as well as the cathode material and potential. A greater discharge capacity can be achieved even in a solvent having a low DN value if the cell is operated such that the cathode potential (U) is within a specific range (i.e., $U > U_{\text{NDR}}$). The operating potential is generally determined by the balance between the applied current density and the external load, the latter of which is affected by various factors, including reaction kinetics and the geometrical structure of the

cathode. Therefore, these parameters should be adjusted accordingly for sustained discharge. This fundamental study of the ORR in Li-O₂ batteries is expected to lead to an improved understanding of the underlying chemistry, which is necessary to realize the development of more sophisticated Li-O₂ units.”

Comment 7: *The authors should give consideration to the effects of viscosity, which may enable a solution reaction acetonitrile even with low solubility, assuming the current is low enough.*

Response: If we understand the Reviewer’s comment correctly, as questioning the effect of viscosity on solubility, we believe that there is no clear correlation between these two parameters. However, if the Reviewer is suggesting that the application of a low current resulted in the predominance of the solution pathway in the acetonitrile electrolyte, then we would refer to the current density applied in the experiments used to generate the data in Fig. 1. As we described in the original manuscript, a relatively high current density of 0.2 mA/cm² was applied to obtain the results in Fig. 1b while employing a carbon paper cathode not having a nano-porous structure (as shown in Fig. 3e). Therefore, it is unlikely that the predominance of the solution pathway in the Fig. 1b data can be attributed to a low current density. It was not our intention to suggest that acetonitrile had a different effect on discharging than other solvents, as we noted in the original manuscript (p. 4, line 30 - p. 6, line 2). Even so, we may not have clearly explained this point, and it is unfortunate that we did not discuss the interaction between the electrode surface and the inhibitor so as to more fully explain the intent of our research. To clarify this point, we have revised the abstract section as follows.

“In non-aqueous Li-O₂ batteries, the one-electron reduction of oxygen and subsequent LiO₂ formation both occur during discharge. This LiO₂ can be converted to insulating Li₂O₂ via two different pathways: a second reduction at the cathode surface or disproportionation in solution. The latter process is known to be advantageous with regard to increasing the discharge capacity and is promoted by a high donor number (DN) electrolyte because of the stability of LiO₂ in media of this type. Herein, we report that the cathodic oxygen reduction reaction during discharge typically exhibits negative differential resistance (NDR). Importantly, we found that the position of the cathode potential relative to the NDR potential region, which varies with the electrolyte, determined the discharge reaction pathway as well as the energy capacity of a Li-O₂ battery. This result implies that the stability of LiO₂ on the cathode also contributes to the determination of the reaction pathway.

”

In addition, we have made an overall revision of the Conclusions section, as we described in the response to Comment 6.

Comment 8: *O₂ reduction in Li acetonitrile does not typically show two peaks and the voltammograms has an appearance of being wet. Further analysis is needed here, i.e. is the first peak quasi-reversible?*

Response: In accordance with the Reviewer’s comment, we confirmed that the second reduction peak in acetonitrile is not due to contamination with water but rather to the adsorption/desorption behavior of the inhibitor species in response to changes in the potential. We obtained CV data using a 3-electrode configuration in a standard electrochemical cell with a glassy carbon electrode, paying careful attention to exclude water contamination. We have added Fig. S7 and a description of this figure to the revised Supplementary Information. The details of the supplementary CV measurements are as follows. Firstly, we acquired CV data at a faster scan rate of 100 mV/s, with the results shown in Figs. S7a to d, and single and double reduction peaks were observed in acetonitrile and DMSO, respectively. These results correspond to values provided in a previous report [Johnson *et al.*, *Nature Chemistry* 6, 1091 (2014)], in which a second reduction peak (at a lower potential) obtained in DMSO was attributed to the formation of Li₂O₂ in the electrolyte, while a single reduction peak was observed in acetonitrile. Moreover, we acquired CV data using acetonitrile within the ORR potential region that corresponds to the linear sweep voltammetry measurements reported by Aetukuri *et al.* [*Nature Chemistry* 7, 50 (2015)]. The data in Figs. S7e and f were generated at scan rates of 10 and 1 mV/s, respectively. Fig. S7e shows a single clear reduction peak resulting from the rapid scan rate. In contrast, two peaks were obtained during cycling at a slower scan rate, as shown in Fig. S7. In addition, the CV data for the second cycle contain a single sharp peak that agrees with the results reported by Aetukuri *et al.* Generally, the adsorption/desorption reactions of inhibitor species, which are determined by the potential, can be clearly detected at very slow scan rates. In contrast, this adsorption/desorption phenomenon is not evident when the scan rate is sufficiently fast relative to the responsiveness of the inhibitor. From these additional results, it can be concluded that the origin of the second reduction peak at 2.3 V obtained at the slower scan rate of 0.5 mV/s (shown in Fig. 4a) was not contamination with water but rather can be ascribed to the adsorption/desorption of the reaction inhibitor on/from the surface of the electrode. However, regrettably, the scan rate used to acquire the data in Fig. 4a was not provided in the original manuscript and therefore these results might be misleading. We have added the necessary details

regarding the experimental conditions (as in the following text) when referencing Fig. 4a in the revised manuscript, on p. 5, line 20.

“It should be emphasized that the CV data in Fig. 4a were obtained at a slow scan rate of 0.5 mV/s, and CV data essentially equivalent to those in the literature were obtained when the scan rate was increased to 100 mV/s (Fig. S7).”

Comment 9: *Voltammetry on a high surface area porous electrode in a cell for cycling is not ideal for electroanalysis. For example, how can the authors confirm that a peak due to depletion of O₂ in the porous structure has not resulted in a peak, followed by further result due to standard linear diffusion to the electrode surface. Moreover, it is suggested that the current does not drop during cycling, even though oxidation is not performance. However, as a high surface area material is used, it is not clear that all the area will be consumed on a cycle. i.e., total surface area may not be limiting the current. This may explain the increase current between d and e on the positive scan. In summary, these data need repeating in a 3-electrode configuration in a standard electrochemical cell at a planer electrode, before these conclusions can be confirmed.*

Response: As the Reviewer noted, the diffusion of O₂ and coverage of the electrodes might have different effects on the CV data acquired from standard planar electrodes and porous structured electrodes. In accordance with this comment, we obtained CV data using a 3-electrode configuration in a standard electrochemical cell with a glassy carbon electrode, making sure to avoid contamination with water, and the additional experimental results have been added to the revised manuscript as Fig. S7. We confirmed that NDR appearing at a slower scan rate can be attributed to the adsorption/desorption of a reaction inhibitor on/from the surface of the electrode. Please see the detailed discussion in response to Comment 8.

As the Reviewer noted, the experiments reported in the original manuscript used a porous electrode, and so all the cathode surface might not have been insulated. However, the incomplete surface coverage of the cathode would not be expected to result in NDR within the ORR potential region. That is, the appearance of NDR is completely unrelated to the electrode structure (i.e., porous or flat). Moreover, the data in Fig. 4a in the original manuscript were acquired using carbon paper as the working electrode at a slower scan rate of 0.5 mV/s, and so it is unlikely that O₂ depletion resulted in the second reduction peak.

Comment 10: *Scan rates and salt concentrations should be added to the legend. Plots should use*

current density. mA cm⁻².

Response: In accordance with the Reviewer's request, we have added information regarding the CV measurements and have corrected the plots in all relevant figures to provide current density values.

Thank you again for your comments on our paper. I trust that the revised manuscript is now suitable for publication.

RESPONSE TO REVIEWER 3:

We wish to express our appreciation to the Reviewer for his or her insightful comments, which have helped us significantly improve the paper.

Comment 1: *The authors claim that higher capacities would be expected when discharge occurs within the NDR region, however the actual data from Figure 4 is somewhat ambiguous on this point. On carbon paper, in both MeCN and DMSO, it appears that all discharge potentials (with the arguable exception of 0.05 mA/cm² in MeCN) occur within the NDR region. It is thus unclear why discharge capacity in MeCN is 3-10x higher than in DMSO. It is also worth noting that the average discharge potential in the DMSO/CP system is higher (2.6-2.8 V) than in MeCN/CP (2.3 - 2.5 V). From several literature reports, it is clear that LiO₂ is more stable at higher potentials, whereas at lower potentials, it would be expected to be easily reduced at the electrode to form Li₂O₂. This expectation is opposite to what the authors find, and miniscule differences in discharge potential relative to the NDR region appear insufficient to account for this.*

Response: We appreciate the Reviewer's comment; however, we respectfully suggest that the reviewer is mistaken on this point. As we described in the section entitled "Origin of NDR and correlation with sustained discharge" in the original manuscript, the discharge capacity is determined by the position of the operating potential (that is, lower or higher) relative to the NDR potential region (the U_{NDR} value). The discharge is sustained over a longer time span as a result of the solution pathway as a result of reduced coverage of adsorbed species when the operating potential is higher than the U_{NDR} . The adsorption isotherm of the adsorbed species changes depending not only on the solvent properties but also on the environment around the adsorbed species, such as the substrate and the structure of the electrode, the temperature and the potential, therefore, U_{NDR} varies with these factors. As a result, employing acetonitrile or DMSO as the electrolyte produces different discharge behaviors in accordance with the difference in the stability of the adsorbed species on each cathode surface. However, as the Reviewer noted, there was some ambiguity in the original manuscript regarding the correlation between NDR and the potential. We also regret that the original explanation of these phenomenon as observed in the CV data in Figs. 4c to 4f in the original manuscript was not appropriate, because the U_{NDR} cannot be accurately obtained from such data, since the observed current is affected by the history of the system during potentiodynamic tests. We therefore conducted additional discharge tests under potentiostatic conditions to assess the U_{NDR} values. We have added the results of potentiostatic discharging trials (as Fig. 5 in the revised manuscript) and a new section entitled "NDR potential region and correlation with sustained discharge" to the revised manuscript (p. 6, line 9 – p. 8, line 10). The results of this additional work demonstrate that the discharge capacity changes significantly the U_{NDR} where the reaction pathway switches. As the Reviewer noted,

the discharge capacity was enhanced in the MeCN system even though the potential obtained with MeCN was lower than that observed using DMSO. These results clearly show that the discharge reaction pathway switches according to the positional relationship between the U_{NDR} value, which is specific for each system, and the discharge potential.

As the Reviewer noted, LiO_2^* has been reported to be readily reduced at the electrode to form Li_2O_2 at lower potentials. However, the pulsed switching of potentials (Fig. 4b) clearly demonstrated that LiO_2^* was stable on the electrode surface and did not transition to Li_2O_2 on a time scale of approximately 10 minutes. The goal of this research was to demonstrate that NDR is an indicator of adsorption/desorption of the inhibitor and to introduce this concept for the first time to the field of Li-O₂ battery research. While it appears that the inhibitor affects the adsorption isotherm of adsorbed species on the cathode, we do not claim that this species is LiO_2^* , as it is beyond the scope of this work to definitively identify the inhibitor. To make this clear to the reader, we have made an overall revision to the Conclusions section and replaced it with the following text.

“We demonstrated that ORR-inhibiting species are generated during the discharge of Li-O₂ batteries, resulting in the appearance of NDR. When the operating potential is higher than the NDR potential region (that is, U_{NDR}), more continuous discharge is obtained as a result of the solution pathway because of the reduced coverage of the cathode by the inhibitor. It is evident that the U_{NDR} value is an important factor determining the continuity of the ORR. Considering the behavior of the inhibitor formed along with the ORR on the cathode and adsorption/desorption in the potential region of the ORR, the inhibiting species is most likely LiO_2^* or a related compound. In principle, the equilibrium potential of Eq. (3), and thus the U_{NDR} value, is determined by the relative stabilities of LiO_2^* and LiO_2^{s} . The stability of LiO_2^{s} is affected by the DN value of the solvent and also by the presence of additives or contaminants, as previously reported, while the stability of LiO_2^* is determined by the electrolyte as well as the cathode material and potential. A greater discharge capacity can be achieved even in a solvent having a low DN value if the cell is operated such that the cathode potential (U) is within a specific range (i.e., $U > U_{\text{NDR}}$). The operating potential is generally determined by the balance between the applied current density and the external load, the latter of which is affected by various factors, including reaction kinetics and the geometrical structure of the cathode. Therefore, these parameters should be adjusted accordingly for sustained discharge. This fundamental study of the ORR in Li-O₂ batteries is expected to lead to an improved understanding of the underlying chemistry, which is necessary to realize the development of more sophisticated Li-O₂ units.”

Comment 2: *The case of Au is particularly intriguing, and reinforces the point from #1. Although*

discharge at 0.02 mA/cm² in the DMSO/Au system results in a higher discharge capacity, its potential is not in the NDR region (as defined by the authors), whereas the discharge potential for MeCN/Au is in the NDR region, but has a lower capacity. The authors acknowledge this inconsistency on lines 6-7 of p. 4, but provide no compelling explanation/hypothesis for the influence of NDR v potential on discharge capacity.

Response: On this point, we feel that we need to clarify some details regarding the work, which may have been misinterpreted. As we discuss in the response to Comment 1, the discharge capacity is determined by the position of the operating potential (either lower or higher) relative to the NDR potential region (that is, the U_{NDR} value). Please see the detailed discussion regarding this comment in the response to Comment 1. The specific discharge behavior obtained with the Au mesh electrode that the Reviewer pointed out can be explained based on the effect of the substrate on the stability of the absorbed species, as in the case of the carbon paper. However, on further reflection, we have decided that it was not appropriate to discuss the relationship between NDR and discharge capacity based on the CV data in Figs. 4c to 4f of the original manuscript. The reasoning behind this reconsideration is as follows.

In the original manuscript, we used carbon paper and Au mesh as the cathode substrates and compared the U_{NDR} values and the discharge properties. Since the surface states, electrode areas and current densities during these measurements were significantly different between the carbon paper and Au mesh trials, it is difficult to accurately estimate the U_{NDR} values and to identify the importance of these values based on the CV data. In the revised manuscript, we have therefore concluded only that NDR generally appears in Li-O₂ battery systems. We have deleted Figs. 1d and 3d, and transferred the results obtained with the Au mesh electrode from the original manuscript (Figs. 4e and 4f) to the revised Supplementary Information (Fig. S8) and added the following text to the revised manuscript (p. 5, line 22).

“The onset of NDR was also confirmed for the DMSO system as well as during trials in which the positive electrode was changed from CP to Au (Fig. S8). These results indicate that NDR is inherent to the positive electrode reaction during the discharge process.”

Comment 3: *Have the authors tried potentiostatically discharging these Li-O₂ cells at potentials chosen to be squarely within/outside the NDR to more unambiguously show its effect on discharge capacity? Such experiments would shed much light on the importance of NDR for Li-O₂ discharge capacity.*

Response: We thank the Reviewer for this insightful comment. In accordance with the Reviewer's comment, we performed additional potentiostatic discharge trials at various potentials. As we discussed in the response to Comment 1, we have added the results of potentiostatic discharging (as Figs. 5 and S12 in the revised manuscript and Supplementary Information, respectively) and have inserted a new section, entitled "NDR potential region and correlation with sustained discharge," into the revised manuscript. These additional results demonstrate the importance of NDR in determining the discharge capacity. Please see the detailed discussion related to this subject in the response to Comment 1.

Comment 4: *The analysis on p. 6 seems to attribute differences in measured current as a function of voltage to differences in the Faradaic current over time, but completely neglects any capacitive effects, which may be important where such low currents (compared to the Faradaic peaks) are recorded. How do the authors know that differences in capacitance (e.g. caused by the potential approaching the potential of zero charge) are not responsible for the reductions in current?*

Response: The capacitive charging current was not included in Fig. 4b due to the low data acquisition frequency. However, upon increasing the data acquisition frequency, the capacitive current can be clearly seen, and so we believe that differences in capacitance are not responsible for the observed variations in current. In the revised manuscript, results acquired using a higher frequency, which demonstrate the capacitive current, are provided as Fig. S14.

Thank you again for your comments on our paper. I trust that the revised manuscript is now suitable for publication.

Reviewers' comments:

Reviewer #1 (Remarks to the Author):

I have carefully gone through the revisions submitted by Hase et al. After the current set of revisions and the changes made, I am of the opinion that the paper has not improved significantly to warrant publication. The following are some critical areas that need improvement/additional data to support the central claims of the paper:

- 1) The NDR window: With the latest set of additions, the figures and the message is totally confusing regards the NDR window. NDR being central to the paper, it is of utmost importance that this be clearly defined. For example, according to Fig. 1b, 2.3 V should be within the NDR region which is in agreement with Fig. 4a. However, according to Fig. 1c 2.3 V is not within the NDR region (albeit for a different cathode) and Fig. 5a. Figs. 5a and 5d show that the NDR is between 1.9-2V.
- 2) The authors seem to argue, based on 6a and 6b that the NDR is a transition region between the surface electrochemical pathway and the solution pathway and essentially implying that surface electrochemical growth of Li₂O₂ would only occur at low potentials. This argument is in direct contradiction with several papers that the authors cited in this manuscript. For example, anhydrous DME has a ORR peak at ~2.5 V, and it has been shown by several researchers that the Li₂O₂ grows as thin films which are passivating.
- 3) The insistence of using LiO₂* as the inhibitor has no evidence anywhere in the paper. While arguing that LiO₂* as an inhibitor is only a hypothesis and that other such inhibitors might exist, the authors have used LiO₂* coverage in Fig. 6a to argue to the existence of an NDR. At this point, it is worth noting that NDR can be attributed to potential-dependent changes in any of the electroactive surface area of the electrode, double layer formation or even the kinetic barriers for charge transfer. In this sense, formation of thin film of Li₂O₂ which is known to be an insulator can also be an equally good and possible inhibitor. It is unclear why the authors have totally ignored Li₂O₂ as an inhibitor and also the possible changes to the double layer structure and/or the kinetic barriers.
- 4) Since NDR is central to the paper and it is this fresh interpretation that might be of considerable import to the field, I suggest that the authors perform additional experiments to support their claims with definitive experimental evidence. For example, current densities in the vicinity of 0.2 mA/cm² that seems to have been used for most of the experiments should also show oscillatory potentials such as those in Fig. 1b or S1b.
- 5) The NDR window in a 2-electrode cell might also depend on the resistance of the electrolyte. In the data presented in the main text, the authors seem to have used an electrolyte with 0.05 M LiTFSI while in the figures in the supplementary the salt concentration seems to be 0.5 M LiTFSI. I would have expected the NDR regions to be different for these two cases, but the data seems to suggest that there is no such electrolyte salt concentration dependence. This needs to be addressed clearly.
- 6) Finally, The authors response to the question on LiOH formation is concerning. In the XRD cell, a leak to air is being posited. How have the authors ensured leak tight electrochemical cells and how have the authors ensured that there is no contamination from the oxygen gas that is bubbled into the Li-O₂ cell. Also, the authors expressed their inability to measure water content in the cells after a measurement. But, this has been routinely measured and there are several papers that have also reported on the formation of water in-situ from organic electrolyte decomposition during the oxidation step in Li-O₂ cells. Therefore, I think that the authors must measure water contents after cell measurements, so as to be certain about the central conclusions of this paper.

Reviewer #2 (Remarks to the Author):

The authors have convincingly dealt with all experimental points raised and have demonstrated an interesting effect. Some theory questions were not answered fully and I remain unconvinced regarding the proposed explanation. Also, the solution mechanism appears inferred from the SEM and capacity rather than any direct measurement. On balance I feel the paper merits publication regardless of my position, but if the authors are to invoke NDR then I do feel they should explain its presence in the system as requested previously.

"In the section "mechanism of the potential Oscillation" the authors clearly describe why the voltage polarises negative. This is already known. They state, "The rapid change of the potential from the negative- to positive-end of the potential oscillation can be explained in the exact same way.". Please also describe in detail the reason for the shift to positive potentials."

Some attempt has been made to do this. The authors added a comprehensive discussion as to why the current oscillates under potentiostatic control and state that the positive shift is due to an increase in surface O₂ concentration and solution resistance. This discussion is logical although I believe there is some error with regards to U, which is stated to be constant and then below changes.

However, when describing the galvanostatic oscillation (that seen in a battery) the negative shift is said to be due to surface adsorption and the positive shift "can be explained in the exactly same way. Namely, the positive potential shift leads to decrease in the coverage of LiO₂* and to increase in the ORR current, leading to the further positive shift of the potential." This explanation is not strong and is contradictory to the above potentiostatic argument and must be clarified before publication.

In summary, I support publication after correction of the above.

Reviewer #3 (Remarks to the Author):

In response to my earlier comments, the authors have clarified that it is the difference between the discharge potential and the NDR potential region that determines the growth mechanism for Li₂O₂: as the potential increases above the NDR region, the solution pathway is promoted and higher discharge capacities are realized. This is a plausible conclusion, however there are a few ambiguities with respect to how the NDR is defined and its universal explanatory power that need to be resolved:

1. The NDR region in MeCN is assumed to be between 1.9 and 2.0 V based on potentiostatic discharge capacities obtained in Figure Figure 5b, however there appears to be no independent assessment of where the NDR should be. Cyclic voltammogram experiments in Figure S8a and c seem to place it between 2.3 and 2.45 V, but this region is not chosen by the authors as the NDR because of the 'dynamic scanning of the potential', although, at 0.5 mV/s, I would expect the system to behave close to its thermodynamic state. Is there a hypothesis for why the NDR obtained from CV is irrelevant for full-cell tests?

2. In DMSO, the authors do not have a particularly convincing explanation for why the capacities below and above the NDR are fairly similar (within error). The explanation that in DMSO, Li₂O₂ always grows via a solution-mediated mechanism contradicts their notion that the NDR acts as a boundary, below which Li₂O₂ grows on the surface, and above which it grows in solution. Thus, it would be

appear that the NDR is irrelevant in DMSO, but relevant for MeCN, and there's no clear explanation why. Moreover, it appears that the authors use the NDR obtained by CV in Figure S8b in assessing the relationship between discharge potential and NDR (which is something they did not do for MeCN). Again, no clear explanation is given for why CV-derived NDR is acceptable for DMSO, but not for MeCN.

Given the observations noted in #2, and the lack of oscillations in discharge curves in DMSO (a telltale sign of NDR as suggested by the authors), it seems to me that the NDR as a boundary between two different growth mechanisms is only relevant in MeCN and less so in DMSO. I would recommend this study for publication if the manuscript were revised to make this emphasis clear, unless otherwise convincing explanations to the contrary can be provided.

RESPONSE TO REVIEWER 1:

Introductory Response: We wish to express our appreciation to the Reviewer for his or her insightful comments, which have helped us significantly improve the paper. In this second resubmission, we have attempted to clarify the focus of the manuscript and address each of the comments. We have also provided an additional explanation regarding the observed fluctuation in the negative differential resistance (NDR) potential region, which was not adequately explained in the previous version of the manuscript.

The aim of this work was to demonstrate that NDR is an indicator of the adsorption/desorption of LiO_2^* (and/or a different ORR inhibitor) and that the NDR potential region represents a zone over which the proportional contributions of the solution and surface pathways transition. We believe that the experimental results provide adequate support for this hypothesis. Figs. 4 and S11 clearly show the presence of an inhibitor that is reversibly absorbed and desorbed according to the potential. As described in the previous version of manuscript (in the section titled “Negative differential resistance”), in the case of an electrochemical reaction system incorporating a reaction inhibitor for which the coverage becomes higher (lower) as E is increased (decreased), NDR will typically be apparent {ref. 48, Krischer, K. *Advances in Electrochemical Sciences and Engineering*, edited by D. M. Kolb, R. C. Alkire (Wiley-VCH, 2003), p. 89}. The results shown in Fig. 5 demonstrate that the NDR potential region represents a boundary at which the dominant reaction pathway changes. To further substantiate our conclusions, we have added a section titled “The characteristic potentiostatic discharge profile associated with the NDR potential region” and Figs. S15 and S16 to the revised Supplementary Information.

The conclusions derived from this work are also supported by the following.

(1) The dominant reaction pathway (either surface or solution) is determined by the $\text{LiO}_2^*/\text{LiO}_2^s$ equilibrium {ref. 29, Johnson, L. *et al.*, *Nat. Chem.* **6**, 1091-1099 (2014)}, (2) the extent of coverage by LiO_2^* varies with the potential and is increased at more negative potentials {ref. 37, Kwabi, D.G. *et al.*, *J. Phys. Chem. Lett.* **7**, 1204-1212 (2016)}, and (3) the adsorbed species LiO_2^* is the most likely candidate for the ORR inhibitor because O_2 must be adsorbed on the active sites of the cathode for the ORR to proceed. Taken together, these points demonstrate that the reversible adsorption/desorption of LiO_2^* based on the potential is the cause of NDR. That is, the proportional contributions of the surface and solution pathways change on the NDR potential region. This conclusion is derived from a logical analysis rather than on our experimental results, but is in good agreement with the experimentally-derived conclusion that we revealed for the first time in this manuscript.

As noted, LiO_2^* is the most probable ORR inhibitor although, unfortunately, we have not specifically confirmed the identity of the inhibitor species using spectroscopic analysis. In the newly

revised manuscript, we have therefore made sure to explain the focus of our work and to point out that the mechanism by which NDR originates at the molecular level is unknown at this point. We have revised the sections entitled “NDR potential region and correlation with sustained discharge,” “Possible origin of NDR and correlation with the reaction pathway” and “Conclusions” to reflect this. The revised parts have been highlighted in the revised manuscript.

We have also revised Fig. 6 to better describe the correlation between the $\text{LiO}_2^*/\text{LiO}_2^s$ ratio and the NDR potential region. The NDR appears in a specific potential region over which the $\text{LiO}_2^*/\text{LiO}_2^s$ ratio drastically changes, as explained using a schematic illustration in Fig. 6a. As in the case of any species adsorbed on an electrode, the characteristics (such as potential area and gradient) of the adsorption isotherm (which reflects the relationship between the $\text{LiO}_2^*/\text{LiO}_2^s$ ratio and the potential in a Li-O₂ battery system) will vary depending on the particular electrode substrate and solvent. Therefore, it is to be expected that each system in our experiments generated a unique adsorption isotherm. As noted in the revised manuscript, the results in Fig. 5 show that the proportional contributions of the solution and surface pathways depend on the potential in the NDR potential region of DMSO system, however, the change slowly and not as sharply as compared with the MeCN system. We reevaluated the experimental results along with the revision of Fig. 6 and the manuscript and have concluded that the result of Fig. S13 was not appropriate to show the change of the proportional contributions of the reaction pathways in the DMSO system. Upon the second resubmission, we deleted the description related to the Fig. S13 and Fig. S13 (in the previous manuscript, p. 8, lines 1-5).

We noted in the previous manuscript that each system produced a unique NDR potential region (p. 7, lines 29–31). However, regrettably, we did not fully explain the relationship between the true electrode potential (E) and the potential controlled and observed in the experimental trials (U), such that fluctuation of the observed NDR potential region (U_{NDR}) was not clearly expressed. Because an ohmic drop (IR , where R is the resistance of the solution between the electrode surface and the reference electrode and I is negative for the reduction current) will be present in all electrochemical reaction systems, U and E will differ according to the equation $U = E + IR$. Under different experimental conditions, IR will have different values and therefore E will vary even if U is constant. As an example, in the case that R has values R_1 and R_2 , the E values will be different (E_1 and E_2) even if the U values are the same, as $U = E_1 + IR_1 = E_2 + IR_2$.

Furthermore, during the discharge of a Li-O₂ battery, Li_2O_2 (which is a low conductivity compound) accumulates and gradually covers the cathode surface, so that the IR value (equal to the difference between the U and E values) changes throughout the discharge process. As a result of these complex interactions, regulated potential conditions (as in Figs. 5b and c) are better suited to the qualitative evaluation of the correlation between the NDR potential region and the discharge capacity, as compared with the combined method performed under dynamic potential (as in Fig. 4a)

and current regulated conditions (as in Fig. 1). We noted the difference between the dynamic potential and regulated conditions in the previous revision of this paper (p. 6, lines 16-18), although we did not describe the effect of continual changes in the IR value. In this second resubmission, we have made a revision to explain more fully (1) that the relationship between U and E is modulated as the apparent NDR potential changes along with the experimental conditions (such as the Li salt concentration and the effective surface area of the cathode) and (2) that the difference between the U and E values changes over time during the Li-O₂ battery discharge. We have therefore revised the section titled “NDR potential region and correlation with sustained discharge.” The revised parts have been highlighted in the revised manuscript.

Comment 1: *The NDR window: With the latest set of additions, the figures and the message is totally confusing regards the NDR window. NDR being central to the paper, it is of utmost importance that this be clearly defined. For example, according to Fig. 1b, 2.3 V should be within the NDR region which is in agreement with Fig. 4a. However, according to Fig. 1c 2.3 V is not within the NDR region (albeit for a different cathode) and Fig. 5a. Figs. 5a and 5d show that the NDR is between 1.9-2V.*

Response: We greatly appreciate the Reviewer’s comment on this point. As the Reviewer noted, we regrettably did not suitably explain that there is a specific and different NDR potential region for each system, and that the observed NDR potential region (U_{NDR}) changes during the discharge process. As described in the Introductory Response section of this document, the IR value is dependent on the particular system employed and also varies over the course of time during discharge. The surface area of the Ketjenblack sheet cathode shown in Fig. 1c is much larger than that of the carbon paper (CP) used in the other experiments, and a different Li salt concentration was employed in each system as well. This variation in the salt concentration directly affects the value of R and indirectly affects the value of I due to the difference in the surface state associated with Li₂O₂ formation. Therefore, it is not unexpected to observe variations in U_{NDR} between Figs. 1, 4, and 5. Moreover, regulated potential conditions (Fig. 5) are more appropriate for the qualitative evaluation of the correlation between NDR potential region and the discharge capacity, as compared with the use of dynamic potential conditions (Figs. 4a and S8). Fig. 4a was included in the manuscript not to specify NDR potential region but rather to indicate that NDR appears in the ORR potential region. Please see our more detailed discussion of this topic in the Introductory Response part of this document.

Comment 2: *The authors seem to argue, based on 6a and 6b that the NDR is a transition region between the surface electrochemical pathway and the solution pathway and essentially implying that surface electrochemical growth of Li₂O₂ would only occur at low potentials. This argument is in direct contradiction with several papers that the authors cited in this manuscript. For example, anhydrous DME has a ORR peak at ~2.5 V, and it has been shown by several researchers that the Li₂O₂ grows as thin films which are passivating.*

Response: On this point, we need to clarify some details regarding this work, which may have been misinterpreted. It was not our intention to suggest that Li₂O₂ is formed only via the surface pathway at lower potentials. In fact, our argument does not contradict these previous reports because each system (MeCN, DMSO, and DME) generated a unique LiO₂* adsorption isotherm that resulted in specific discharge properties. Moreover, the results obtained in this work demonstrate that the adsorption isotherms changed abruptly over a narrow potential region in MeCN whereas this transition was slower and less clear in DMSO. Please see the revised Fig. 6 and the detailed discussion in the Introductory Response part of this document.

Comment 3: *The insistence of using LiO₂* as the inhibitor has no evidence anywhere in the paper. While arguing that LiO₂* as an inhibitor is only a hypothesis and that other such inhibitors might exist, the authors have used LiO₂* coverage in Fig. 6a to argue to the existence of an NDR. At this point, it is worth noting that NDR can be attributed to potential-dependent changes in any of the electroactive surface area of the electrode, double layer formation or even the kinetic barriers for charge transfer. In this sense, formation of thin film of Li₂O₂ which is known to be an insulator can also be an equally good and possible inhibitor. It is unclear why the authors have totally ignored Li₂O₂ as an inhibitor and also the possible changes to the double layer structure and/or the kinetic barriers.*

Response: As we discuss in the Introductory Response portion of this document, we believe that LiO₂* is the most likely candidate for the inhibitor. Please see the detailed discussion in the Introductory Response portion. However, because the Reviewer has noted the possibility that the formation of an electric double layer or additional effects other than the potential dependence of reaction inhibitor coverage could produce NDR, we have addressed these points below.

The possibility of an electric double layer effect can be ruled out for the following reasons. As shown in Fig. 1 attached at the end of this response, a potential region exists in the electric double layer over which the reduction current (oxidation current) decreases (increases) as the potential is swept toward negative (positive) values. The current responses shown in the attached Fig. 1 simply

result from charging and discharging of the electric double layer, and are different in principle from NDR. In fact, it can be demonstrated mathematically that electrochemical oscillation will not result from the charge/discharge currents of an electric double layer. Moreover, the current responses shown in Figs. 5a and S8 could not have been derived from an electric double layer since these CV data were obtained at a suitably slow scan rate of 0.5 mV/s. Under these conditions, the discharge and charge currents of the electric double layer will not be reflected in the CV curves acquired at a current density of several $\mu\text{A}/\text{cm}^2$. We have added Fig. S15 to the revised Supplementary Information to present data obtained acquired using the same conditions as in Fig. 4a but under an Ar atmosphere. Fig. S15 confirms that the discharge and charge currents in the electric double layer are not included in the CV curves acquired under these conditions. Furthermore, Fig. S14 demonstrates that the time constants of the charge and discharge of the electric double layer are much smaller than that of NDR.

As an aside, we note that it is important to consider the current originating from electrolyte decomposition, as shown in Fig. S15. During galvanostatic discharge measurements, the effect of electrolyte decomposition is negligibly small because the decomposition potential is lower than the ORR potential. However, the decomposition current has a slight effect on the results obtained during potentiostatic and CV measurements at potentials lower than 2.4 V.

With regard to other effects, as we demonstrate via equation (6) in the main manuscript, there are two possible causes for the appearance of NDR. These are $dA/dE < 0$ or $dk/dE < 0$, indicating that NDR is induced by kinetic barriers, as discussed in the section titled “Possible origin of NDR and correlation with the reaction pathway” in the previous manuscript. The former appears when the coverage of the inhibitor increases along with a negative potential shift (that is, an increase in the overpotential), whereas the latter appears as the coverage of the promoter decreases. In the former case, the NDR appears due to the suppression of the reaction on the negative potential side, while in the latter case the NDR appears as the reaction rate (which is enhanced on the positive potential side) returns to normal on the negative potential side. In this work, we determined that $dA/dk < 0$ (that is, the inhibitor type) is more likely to occur in this system, because the current is greatly suppressed when the potential is swept in the positive direction (c-d-e) as shown in the CV data in Fig. 4a.

Comment 4: *Since NDR is central to the paper and it is this fresh interpretation that might be of considerable import to the field, I suggest that the authors perform additional experiments to support their claims with definitive experimental evidence. For example, current densities in the vicinity of 0.2 mA/cm² that seems to have been used for most of the experiments should also show oscillatory potentials such as those in Fig. 1b or S1b.*

Response: If we understand the Reviewer's comment correctly, "*their claim*" in the first sentence of the comment is the existence of NDR in the ORR potential region. As such, we maintain that we have provided direct evidence for the appearance of NDR in Figs. 4 and S8 of the previous manuscript. Moreover, because the appearance of electrochemical oscillation is an essential prerequisite for NDR, Fig. 1b also demonstrates the appearance of NDR (ref. 48). We interpreted the sentence "...current densities in the vicinity of 0.2 mA/cm² that seems to have been used for most of the experiments should also show oscillatory potentials..." in the comment to mean "...the current of approximately 0.2 mA/cm² obtained during the potentiostatic experiments should show current oscillations...". In this regard, we have already demonstrated via Fig. S9 that current oscillations were clearly generated in the vicinity of a current density of 0.2 mA/cm² at potentials of 2.2 V and 2.4 V.

Comment 5: *The NDR window in a 2-electrode cell might also depend on the resistance of the electrolyte. In the data presented in the main text, the authors seem to have used an electrolyte with 0.05 M LiTFSI while in the figures in the supplementary the salt concentration seems to be 0.5 M LiTFSI. I would have expected the NDR regions to be different for these two cases, but the data seems to suggest that there is no such electrolyte salt concentration dependence. This needs to be addressed clearly.*

Response: As the reviewer noted, the observed NDR potential region (U_{NDR}) is changed in accordance with the salt concentration. As we describe in the Introductory Response part of this document, we do not claim that the appearance and characteristics of NDR will be universal regardless of the conditions. Moreover, it is not appropriate to discuss the effect of the salt concentration on U_{NDR} with comparing the results shown in Fig. 4a and Fig.5 b, because experimental methods between them were different. Please see the detailed discussion in the Introductory Response part and the Response to the Comment 1 associated with the effect of the salt concentration on U_{NDR} .

In addition, as we described in the previous manuscript, the results shown in Fig. S9 (obtained in a 0.5 M LiTFSI/MeCN electrolyte) were not considered suitable for an accurate estimation of the NDR potential region because the current oscillation appeared under these conditions (p.6, lines 19 - 23).

Comment 6: *Finally, The authors response to the question on LiOH formation is concerning. In the XRD cell, a leak to air is being posited. How have the authors ensured leak tight electrochemical*

cells and how have the authors ensured that there is no contamination from the oxygen gas that is bubbled into the Li-O₂ cell. Also, the authors expressed their inability to measure water content in the cells after a measurement. But, this has been routinely measured and there are several papers that have also reported on the formation of water *in-situ* from organic electrolyte decomposition during the oxidation step in Li-O₂ cells. Therefore, I think that the authors must measure water contents after cell measurements, so as to be certain about the central conclusions of this paper

Response: In accordance with the Reviewer's comment, we repeated the PXRD analyses, paying careful attention to exclude any potential contamination with water. We have revised Fig. 2b to include the new data, in which reflections originating from LiOH are not observed. We have also included an expanded spectra in Fig. 2b that extends to smaller angles at the end of this document, as Fig. 2. Neither of the potential LiOH peaks at 20.5° (0 0 1) and 32.6° (1 0 1) are present in the revised spectrum. Therefore, we conclude that a quantity of LiOH sufficient for detection by PXRD was not formed as a discharge product on the cathode. Together with the discussion in the section titled "Views on the possible effects of contamination with water" provided in the previous Supplementary Information, these results clearly address any concerns regarding the effects of contamination with water and formation of LiOH as a factor affecting our experimental results.

The Reviewer also states that "... *the authors expressed their inability to measure water content in the cells after a measurement. But, this has been routinely measured ...*". Regrettably, we have been unable to find reports regarding the assessment of water content after battery tests. In addition, as we stated in our response to Reviewer #2 in the previous "Response to the Reviewers" document, the small amount of electrolyte used in our experiments (200 μ L per cell, as noted in the Supplementary Information) made it difficult to accurately determine low water concentrations (less than 30 ppm) by Karl Fischer titration. We would greatly appreciate it if the Reviewer can provide information regarding the specific manuscripts in which the water analysis procedure performed after the battery tests is described.

The Reviewer also pointed out the possible formation of water *in situ* via decomposition of the organic electrolyte during the oxidation step in Li-O₂ cells. Chen *et al.* reported that water was formed by decomposition of the electrolyte, beginning with a nucleophilic attack by an O₂ anion radical {*J. Am. Chem. Soc.* **134**, 7952-7957 (2012)}. If this nucleophilic attack is towards MeCN, the partially positive carbon of the nitrile group would be the most likely target {Mizuno, F. *et al. Electrochemistry* **79**, 876-881 (2011)}. However, since acetonitrile does not have a β -hydrogen on the nitrile group, it cannot generate a 6-membered ring transition state associated with a carbon radical intermediate, as suggested by Chen *et al.* Therefore, it is unlikely that water is produced via the decomposition of acetonitrile in our system. It has also been demonstrated experimentally that acetonitrile is stable in the presence of O₂ anion radicals {Takechi, K. *et al. ECS Electrochem. Lett.*

1, A27-A29 (2012)}.

As we discuss in the Introductory Response part to this document, the primary aim of this manuscript was to demonstrate that NDR is an indicator of the adsorption/desorption of LiO_2^* , and that the NDR potential region value represents the region over which the reaction pathway changes. The conclusions related to these two topics are valid regardless of whether LiOH is formed on the cathode or in the electrolyte. Furthermore, as we previously described in the Manuscript and Supplementary Information (in the sections titled “NDR potential region and correlation with sustained discharge” and “Views on the possible effects of contamination with water”), the discharge capacity obtained in the MeCN system within the potential region more negative than the NDR potential region is in good agreement with the theoretical value estimated for a reaction proceeding solely via the surface pathway. The above results demonstrate that, even if water was present in our electrochemical measurement system, this would have had a minimal effect on the main conclusions of the present work.

Thank you again for your comments on our paper. We trust that the revised manuscript is now suitable for publication.

Attached Fig. 1 | Schematic illustration for the possible difference of I-V curves between the oxidation/reduction current with double layer formation and NDR phenomena.

Attached Fig. 2 | The *ex-situ* PXRD patterns. PXRD patterns of the CP cathode were obtained before (a), after discharging in the MeCN electrolyte (same spectrum of the revised Fig. 2b) (b). The curves are normalized to the CP's (002) XRD peak. These results indicate that Li_2O_2 formed on the cathode was reactive with water and the content of LiOH in the sample increased with time. Both peaks of LiOH at 20.5° (0 0 1) and 32.6° (1 0 1) were not detected in the revised spectrum.

RESPONSE TO REVIEWER 2:

Introductory Response: We wish to express our appreciation to the Reviewer for his or her insightful comments, which have helped us significantly improve the paper. In this second resubmission, we have attempted to clarify the focus of the manuscript and address each of the comments. We have also provided an additional explanation regarding the observed fluctuation in the negative differential resistance (NDR) potential region, which was not adequately explained in the previous version of the manuscript.

The aim of this work was to demonstrate that NDR is an indicator of the adsorption/desorption of LiO_2^* (and/or a different ORR inhibitor) and that the NDR potential region represents a zone over which the proportional contributions of the solution and surface pathways transition. We believe that the experimental results provide adequate support for this hypothesis. Figs. 4 and S11 clearly show the presence of an inhibitor that is reversibly absorbed and desorbed according to the potential. As described in the previous version of manuscript (in the section titled “Negative differential resistance”), in the case of an electrochemical reaction system incorporating a reaction inhibitor for which the coverage becomes higher (lower) as E is increased (decreased), NDR will typically be apparent {ref. 48, Krischer, K. *Advances in Electrochemical Sciences and Engineering*, edited by D. M. Kolb, R. C. Alkire (Wiley-VCH, 2003), p. 89}. The results shown in Fig. 5 demonstrate that the NDR potential region represents a boundary at which the dominant reaction pathway changes. To further substantiate our conclusions, we have added a section titled “The characteristic potentiostatic discharge profile associated with the NDR potential region” and Figs. S15 and S16 to the revised Supplementary Information.

The conclusions derived from this work are also supported by the following.

(1) The dominant reaction pathway (either surface or solution) is determined by the $\text{LiO}_2^*/\text{LiO}_2^s$ equilibrium {ref. 29, Johnson, L. *et al.*, *Nat. Chem.* **6**, 1091-1099 (2014)}, (2) the extent of coverage by LiO_2^* varies with the potential and is increased at more negative potentials {ref. 37, Kwabi, D.G. *et al.*, *J. Phys. Chem. Lett.* **7**, 1204-1212 (2016)}, and (3) the adsorbed species LiO_2^* is the most likely candidate for the ORR inhibitor because O_2 must be adsorbed on the active sites of the cathode for the ORR to proceed. Taken together, these points demonstrate that the reversible adsorption/desorption of LiO_2^* based on the potential is the cause of NDR. That is, the proportional contributions of the surface and solution pathways change on the NDR potential region. This conclusion is derived from a logical analysis rather than on our experimental results, but is in good agreement with the experimentally-derived conclusion that we revealed for the first time in this manuscript.

As noted, LiO_2^* is the most probable ORR inhibitor although, unfortunately, we have not specifically confirmed the identity of the inhibitor species using spectroscopic analysis. In the newly

revised manuscript, we have therefore made sure to explain the focus of our work and to point out that the mechanism by which NDR originates at the molecular level is unknown at this point. We have revised the sections entitled “NDR potential region and correlation with sustained discharge,” “Possible origin of NDR and correlation with the reaction pathway” and “Conclusions” to reflect this. The revised parts have been highlighted in the revised manuscript.

We have also revised Fig. 6 to better describe the correlation between the $\text{LiO}_2^*/\text{LiO}_2^s$ ratio and the NDR potential region. The NDR appears in a specific potential region over which the $\text{LiO}_2^*/\text{LiO}_2^s$ ratio drastically changes, as explained using a schematic illustration in Fig. 6a. As in the case of any species adsorbed on an electrode, the characteristics (such as potential area and gradient) of the adsorption isotherm (which reflects the relationship between the $\text{LiO}_2^*/\text{LiO}_2^s$ ratio and the potential in a Li-O₂ battery system) will vary depending on the particular electrode substrate and solvent. Therefore, it is to be expected that each system in our experiments generated a unique adsorption isotherm. As noted in the revised manuscript, the results in Fig. 5 show that the proportional contributions of the solution and surface pathways depend on the potential in the NDR potential region of DMSO system, however, the change slowly and not as sharply as compared with the MeCN system. We reevaluated the experimental results along with the revision of Fig. 6 and the manuscript and have concluded that the result of Fig. S13 was not appropriate to show the change of the proportional contributions of the reaction pathways in the DMSO system. Upon the second resubmission, we deleted the description related to the Fig. S13 and Fig. S13 (in the previous manuscript, p. 8, lines 1-5).

We noted in the previous manuscript that each system produced a unique NDR potential region (p. 7, lines 29–31). However, regrettably, we did not fully explain the relationship between the true electrode potential (E) and the potential controlled and observed in the experimental trials (U), such that fluctuation of the observed NDR potential region (U_{NDR}) was not clearly expressed. Because an ohmic drop (IR , where R is the resistance of the solution between the electrode surface and the reference electrode and I is negative for the reduction current) will be present in all electrochemical reaction systems, U and E will differ according to the equation $U = E + IR$. Under different experimental conditions, IR will have different values and therefore E will vary even if U is constant. As an example, in the case that R has values R_1 and R_2 , the E values will be different (E_1 and E_2) even if the U values are the same, as $U = E_1 + IR_1 = E_2 + IR_2$.

Furthermore, during the discharge of a Li-O₂ battery, Li_2O_2 (which is a low conductivity compound) accumulates and gradually covers the cathode surface, so that the IR value (equal to the difference between the U and E values) changes throughout the discharge process. As a result of these complex interactions, regulated potential conditions (as in Figs. 5b and c) are better suited to the qualitative evaluation of the correlation between the NDR potential region and the discharge capacity, as compared with the combined method performed under dynamic potential (as in Fig. 4a)

and current regulated conditions (as in Fig. 1). We noted the difference between the dynamic potential and regulated conditions in the previous revision of this paper (p. 6, lines 16-18), although we did not describe the effect of continual changes in the IR value. In this second resubmission, we have made a revision to explain more fully (1) that the relationship between U and E is modulated as the apparent NDR potential changes along with the experimental conditions (such as the Li salt concentration and the effective surface area of the cathode) and (2) that the difference between the U and E values changes over time during the Li-O₂ battery discharge. We have therefore revised the section titled “NDR potential region and correlation with sustained discharge.” The revised parts have been highlighted in the revised manuscript.

Comment 1: *The authors have convincingly dealt with all experimental points raised and have demonstrated an interesting effect. Some theory questions were not answered fully and I remain unconvinced regarding the proposed explanation. Also, the solution mechanism appears inferred from the SEM and capacity rather than any direct measurement. On balance I feel the paper merits publication regardless of my position, but if the authors are to invoke NDR then I do feel they should explain its presence in the system as requested previously.*

Response: We greatly appreciate the Reviewer’s comment on this point. As we described in the Introductory Response part to this document, the primary goal of this manuscript was to demonstrate that NDR is an indicator of the adsorption/desorption of LiO₂* (and/or another ORR inhibitor) and that the NDR potential region represents a transition between the solution and surface pathways. We believe that the experimental data and a logical analysis of the results adequately support our conclusions with regard to these subjects. Please see the detailed discussion in the Introductory Response part. However, as the Reviewer noted, we determined the proposed reaction pathway based on the discharge capacity data and SEM images and have not identified the presence of LiO₂* or LiO₂^s or their relative ratios using electrochemical or spectroscopic methods. We hope to address this aspect of the work in future.

However, at present, we maintain that we have demonstrated the existence of NDR based on the direct evidence provided in Figs. 4 and S8. It should be noted that Fig. 1b also confirms the appearance of NDR, as an observation of electrochemical oscillation is an essential prerequisite for this phenomenon {ref. 48, Krischer, K. *Advances in Electrochemical Sciences and Engineering*, edited by D. M. Kolb, R. C. Alkire (Wiley-VCH, 2003), p.89}.

Comment 2: *Some attempt has been made to do this. The authors added a comprehensive*

discussion as to why the current oscillates under potentiostatic control and state the that positive shift is due to an increase in surface O₂ concentration and solution resistance. This discussion is logical although I believe there is some error with regards to U, which is stated to be constant and then below changes.

However, when describing the galvanostatic oscillation (that seen in a battery) the negative shift is said to be due to surface adsorption and the positive shift “can be explained in the exactly same way. Namely, the positive potential shift leads to decrease in the coverage of LiO₂ and to increase in the ORR current, leading to the further positive shift of the potential.” This explanation is not strong and is contradictory to the above potentiostatic argument and must be clarified before publication.*

Response: As the Reviewer noted, our previous discussion may have been confusing. We also regret that the part of the discussion in the section titled “Mechanism of the potential oscillation” in the previous Supplementary Information was incorrect. Therefore, we have fully revised this section. In the newly revised Supplementary Information, the revised parts have been highlighted.

Thank you again for your comments on our paper. We trust that the revised manuscript is now suitable for publication.

RESPONSE TO REVIEWER 3:

Introductory Response: We wish to express our appreciation to the Reviewer for his or her insightful comments, which have helped us significantly improve the paper. In this second resubmission, we have attempted to clarify the focus of the manuscript and address each of the comments. We have also provided an additional explanation regarding the observed fluctuation in the negative differential resistance (NDR) potential region, which was not adequately explained in the previous version of the manuscript.

The aim of this work was to demonstrate that NDR is an indicator of the adsorption/desorption of LiO_2^* (and/or a different ORR inhibitor) and that the NDR potential region represents a zone over which the proportional contributions of the solution and surface pathways transition. We believe that the experimental results provide adequate support for this hypothesis. Figs. 4 and S11 clearly show the presence of an inhibitor that is reversibly absorbed and desorbed according to the potential. As described in the previous version of manuscript (in the section titled “Negative differential resistance”), in the case of an electrochemical reaction system incorporating a reaction inhibitor for which the coverage becomes higher (lower) as E is increased (decreased), NDR will typically be apparent {ref. 48, Krischer, K. *Advances in Electrochemical Sciences and Engineering*, edited by D. M. Kolb, R. C. Alkire (Wiley-VCH, 2003), p. 89}. The results shown in Fig. 5 demonstrate that the NDR potential region represents a boundary at which the dominant reaction pathway changes. To further substantiate our conclusions, we have added a section titled “The characteristic potentiostatic discharge profile associated with the NDR potential region” and Figs. S15 and S16 to the revised Supplementary Information.

The conclusions derived from this work are also supported by the following.

(1) The dominant reaction pathway (either surface or solution) is determined by the $\text{LiO}_2^*/\text{LiO}_2^s$ equilibrium {ref. 29, Johnson, L. *et al.*, *Nat. Chem.* **6**, 1091-1099 (2014)}, (2) the extent of coverage by LiO_2^* varies with the potential and is increased at more negative potentials {ref. 37, Kwabi, D.G. *et al.*, *J. Phys. Chem. Lett.* **7**, 1204-1212 (2016)}, and (3) the adsorbed species LiO_2^* is the most likely candidate for the ORR inhibitor because O_2 must be adsorbed on the active sites of the cathode for the ORR to proceed. Taken together, these points demonstrate that the reversible adsorption/desorption of LiO_2^* based on the potential is the cause of NDR. That is, the proportional contributions of the surface and solution pathways change on the NDR potential region. This conclusion is derived from a logical analysis rather than on our experimental results, but is in good agreement with the experimentally-derived conclusion that we revealed for the first time in this manuscript.

As noted, LiO_2^* is the most probable ORR inhibitor although, unfortunately, we have not specifically confirmed the identity of the inhibitor species using spectroscopic analysis. In the newly

revised manuscript, we have therefore made sure to explain the focus of our work and to point out that the mechanism by which NDR originates at the molecular level is unknown at this point. We have revised the sections entitled “NDR potential region and correlation with sustained discharge,” “Possible origin of NDR and correlation with the reaction pathway” and “Conclusions” to reflect this. The revised parts have been highlighted in the revised manuscript.

We have also revised Fig. 6 to better describe the correlation between the $\text{LiO}_2^*/\text{LiO}_2^s$ ratio and the NDR potential region. The NDR appears in a specific potential region over which the $\text{LiO}_2^*/\text{LiO}_2^s$ ratio drastically changes, as explained using a schematic illustration in Fig. 6a. As in the case of any species adsorbed on an electrode, the characteristics (such as potential area and gradient) of the adsorption isotherm (which reflects the relationship between the $\text{LiO}_2^*/\text{LiO}_2^s$ ratio and the potential in a Li-O₂ battery system) will vary depending on the particular electrode substrate and solvent. Therefore, it is to be expected that each system in our experiments generated a unique adsorption isotherm. As noted in the revised manuscript, the results in Fig. 5 show that the proportional contributions of the solution and surface pathways depend on the potential in the NDR potential region of DMSO system, however, the change slowly and not as sharply as compared with the MeCN system. We reevaluated the experimental results along with the revision of Fig. 6 and the manuscript and have concluded that the result of Fig. S13 was not appropriate to show the change of the proportional contributions of the reaction pathways in the DMSO system. Upon the second resubmission, we deleted the description related to the Fig. S13 and Fig. S13 (in the previous manuscript, p. 8, lines 1-5).

We noted in the previous manuscript that each system produced a unique NDR potential region (p. 7, lines 29–31). However, regrettably, we did not fully explain the relationship between the true electrode potential (E) and the potential controlled and observed in the experimental trials (U), such that fluctuation of the observed NDR potential region (U_{NDR}) was not clearly expressed. Because an ohmic drop (IR , where R is the resistance of the solution between the electrode surface and the reference electrode and I is negative for the reduction current) will be present in all electrochemical reaction systems, U and E will differ according to the equation $U = E + IR$. Under different experimental conditions, IR will have different values and therefore E will vary even if U is constant. As an example, in the case that R has values R_1 and R_2 , the E values will be different (E_1 and E_2) even if the U values are the same, as $U = E_1 + IR_1 = E_2 + IR_2$.

Furthermore, during the discharge of a Li-O₂ battery, Li_2O_2 (which is a low conductivity compound) accumulates and gradually covers the cathode surface, so that the IR value (equal to the difference between the U and E values) changes throughout the discharge process. As a result of these complex interactions, regulated potential conditions (as in Figs. 5b and c) are better suited to the qualitative evaluation of the correlation between the NDR potential region and the discharge capacity, as compared with the combined method performed under dynamic potential (as in Fig. 4a)

and current regulated conditions (as in Fig. 1). We noted the difference between the dynamic potential and regulated conditions in the previous revision of this paper (p. 6, lines 16-18), although we did not describe the effect of continual changes in the IR value. In this second resubmission, we have made a revision to explain more fully (1) that the relationship between U and E is modulated as the apparent NDR potential changes along with the experimental conditions (such as the Li salt concentration and the effective surface area of the cathode) and (2) that the difference between the U and E values changes over time during the Li-O₂ battery discharge. We have therefore revised the section titled “NDR potential region and correlation with sustained discharge.” The revised parts have been highlighted in the revised manuscript.

Comment 1: *The NDR region in MeCN is assumed to be between 1.9 and 2.0 V based on potentiostatic discharge capacities obtained in Figure Figure 5b, however there are appears to be no independent assessment of where the NDR should be. Cyclic voltammogram experiments in Figure S8a and c seem to place it between 2.3 and 2.45 V, but this region is not chosen by the authors as the NDR because of the 'dynamic scanning of the potential', although, at 0.5 mV/s, I would expect the system to behave close to its thermodynamic state. Is there a hypothesis for why the NDR obtained fro CV is irrelevant for full-cell tests?*

Response: We greatly appreciate the Reviewer’s comment. As we described in the Introductory Response part of this document, regulated potential conditions (in which U is fixed at a certain value) are better suited to the qualitative evaluation of the correlation between U and the discharge capacity. As the Reviewer noted, the scan rate in Figs. 4a and S8 is relatively slow at 0.5 mV/s. However, as shown in Fig. 4a, the positive and negative curves obtained during each sweep are not overlapped even at this slow scan rate. This result clearly implies that the sweep rate of 0.5 mV/s was still too rapid to allow an evaluation of the stationary thermodynamic states of these systems.

As the reviewer noted, an independent assessment of the NDR potential region would be beneficial in terms of fully explaining this phenomenon. Computational chemistry techniques are one alternative strategy to elucidating the independent adsorption isotherms of the inhibitor and NDR potential regions. However, the identity of the inhibitor must be ascertained before these calculations can be performed, and we have not yet identified the molecular species using spectroscopic methods. Please see the detailed discussion in the Introductory Response part. The primary goal of this work was to demonstrate that NDR is an indicator of the adsorption/desorption of LiO₂* and that the NDR potential region represents the transition between the reaction pathways. We feel that these conclusions are adequately supported experimentally as well as logically, although we concur with the necessity of an independent assessment. In the newly revised Supplementary

Information, we have added further discussion based on the results of potentiostatic discharge trials (Fig. 5b). These data show that the NDR potential region represents a boundary at which the dominant reaction pathway is changed. Please see the detailed discussion in the section titled “The characteristic potentiostatic discharge profile associated with the NDR potential region” and Figs. S15 and S16 in the revised Supplementary Information.

Comment 2: *In DMSO, the authors do not have a particularly convincing explanation for why the capacities below and above the NDR are fairly similar (within error). The explanation that in DMSO, Li₂O₂ always grows via a solution-mediated mechanism contradicts their notion that the NDR acts as a boundary, below which Li₂O₂ grows on the surface, and above which it grows in solution. Thus, it would appear that the NDR is irrelevant in DMSO, but relevant for MeCN, and there's no clear explanation why. Moreover, it appears that the authors use the NDR obtained by CV in Figure S8b in assessing the relationship between discharge potential and NDR (which is something they did not do for MeCN). Again, no clear explanation is given for why CV-derived NDR is acceptable for DMSO, but not for MeCN.*

Response: As the reviewer noted, Fig. 6 of the previous manuscript regrettably gave the impression that Li₂O₂ was formed solely via the solution pathway (or the surface pathway) at potentials higher (or lower) than the NDR potential. As we described in the Introductory Response part to this document, we have revised Fig. 6 to better describe the correlation between the extent of coverage by the inhibitor and the NDR potential region. The characteristics of the inhibitor adsorption isotherm, which represents the LiO₂^{*}/LiO₂^s ratio as a function of the potential, will change depending on the particular electrode substrate and solvent employed. Therefore, each system will generate a unique adsorption isotherm. The results obtained in this work demonstrate that the adsorption isotherms change abruptly within a narrow potential region in MeCN but change slowly and not as sharply in DMSO. Both the solution and surface pathways always proceed in the ORR potential region but the ratio between these mechanisms varies with the specific adsorption isotherm for each system. We have revised Fig. 6 as discussed in the Introductory Response part. The revised explanation is consistent with the experimental results as well as with previous reports.

Fig. S8 provides direct evidence for the presence of NDR in the ORR potential region of Li-O₂ batteries. In the manuscript, we do not state that NDR determined by CV is acceptable for DMSO but not for MeCN. Fig. S8 was included not to specify the NDR potential region but to indicate that NDR generally appears in the ORR potential region of a Li-O₂ battery regardless of the system. Please see the detailed discussion in the Introductory Response part.

Comment in the last paragraph: *Given the observations noted in #2, and the lack of oscillations in discharge curves in DMSO (a telltale sign of NDR as suggested by the authors), it seems to me that the NDR as a boundary between two different growth mechanisms is only relevant in MeCN and less so in DMSO. I would recommend this study for publication if the manuscript were revised to make this emphasis clear, unless otherwise convincing explanations to the contrary can be provided.*

Response: As we state in our response to comment 2, NDR typically appears in the ORR potential region of Li-O₂ batteries. The potential region of negative resistance ($dI/dU < 0$) shown in Fig. S8 provides evidence for the existence of NDR in the DMSO systems. However, the contribution of NDR to the proportional occurrence of the solution and surface pathways is dependent on the particular system because each system generates a unique adsorption isotherm. As we described in the Response to comment 2, the adsorption isotherms change abruptly within a narrow potential region in MeCN but change slowly and not as sharply in DMSO.

Typically, a higher NDR results in a stronger feedback mechanism, as described in the Supplementary Information section of the previous manuscript. The stronger feedback mechanism renders the system unstable, leading to spontaneous electrochemical oscillation (ref. 48). Therefore, this oscillation may not be observed in the case that the degree of NDR is relatively small, as is evident for the DMSO system (Figs. 5c and 5d). We discussed the mechanism by which electrochemical oscillation appears only in the MeCN system in the previous manuscript (p. 8, lines 4-8). Moreover, the NDR potential region of DMSO system also represents a boundary at which the proportional contributions of the solution and surface pathways are changed. In the newly revised Supplementary Information, we have added the discussion based on the results of DMSO system (Fig. 5c). Please see the detailed discussion in the section titled “The characteristic potentiostatic discharge profile associated with the NDR potential region” and Figs. S15 and S16 in the revised Supplementary Information.

Thank you again for your comments on our paper. We trust that the revised manuscript is now suitable for publication.

Reviewers' comments:

Reviewer #1 (Remarks to the Author):

I am satisfied with the changes made to the manuscript and the response to the queries that I had during the previous iteration. I recommend publication of this manuscript.

Reviewer #2 (Remarks to the Author):

Unfortunately, the authors have not responded to my single request (see below) and therefore I cannot support publication;

.... when describing the galvanostatic oscillation (that seen in a battery) the negative shift is said to be due to surface adsorption and the positive shift "can be explained in the exactly same way. Namely, the positive potential shift leads to decrease in the coverage of LiO_2^* and to increase in the ORR current, leading to the further positive shift of the potential." This explanation is not strong and is contradictory

For clarity, the current oscillation mechanism is clear and the influence of the IR drop logical. But during galvanostatic measurements, no argument has been presented to explain why the "positive feedback", resulting in a negative shift in potential, stops and shifts in the opposite positive direction. My expectation is that this will simply continue to shift negative and reduce solvent. The authors state "The rapid change of the potential from the negative- to positive-end of the potential oscillation can be explained in exactly the same way." This is not sufficient and needs further explanation.

They only state that

"the positive potential shift leads to a decrease in coverage by LiO_2^* and to an increase in the ORR current, producing an additional positive shift of the potential.

However above state that LiO_2 coverage results in a negative shift.

"in coverage by the LiO_2^* (i.e., the inhibitor) and so decreases the ORR current. This, in turn, results in a further negative shift of the potential, in a positive-feedback manner, to maintain a constant current."

In summary, there has to be a trigger to drive the positive shift and I do not believe surface coverage is the reason. If the authors are unable to explain this then I would have to conclude that their theory is incorrect.

Reviewer #3 (Remarks to the Author):

My major concern with this study has to do with the conclusion that the NDR represents a boundary region separating surface from solution phase-mediated Li_2O_2 electrodeposition. My critique was that in DMSO, NDR seems to have little effect: the discharge capacities obtained below and above the NDR are quite similar, and no voltage oscillations characteristic of the NDR are obtained in discharge experiments in DMSO.

The authors respond by stressing that: 1. Figure S8b and d demonstrate an NDR region where the absolute value of the current increases within the capacitive region while the potential is swept in a more positive direction and more importantly, that 2. the NDR more directly describes the adsorption of the ORR inhibitor, but not the extent of coverage given by the adsorption isotherm. Thus, in DMSO, there is a much smoother transition between the surface and solution pathways, leading to the more gradual increase in capacity shown in Figure 5d for DMSO. This qualification is important because although several questions about the NDR remain unresolved (its relative absence from previous Li-O₂ discharge studies with similar experimental setups, the identity of the inhibitor, why no oscillations are observed in DMSO, why a solution-mediated mechanism is proposed to occur at as low a potential as 2.0 V in MeCN, in contrast to other literature) it better rationalizes how observations in MeCN and DMSO belong within the same conceptual framework.

RESPONSE TO REVIEWER 2:

Introductory Response: We wish to express our great appreciation to the Reviewer for his or her insightful comments, which have helped us significantly improve the paper. In advance of the response to the comment, we would like to mention the major point of this revision.

The experimental results added to the revised manuscript unveiled the common framework to be applied to Li-O₂ batteries. Meanwhile, we maintain our argument and conclusion which have been discussed in our previous version of manuscripts. The framework, which was additionally shown in the newly revised manuscript, clearly explains both of our results and previous reports in a comprehensive manner. In order to clarify the framework upon this resubmission, additional experiments employing Au electrode were performed and the whole of the manuscript was revised. The outline of the framework is described as follows.

In the previous version of manuscript, we clarified the framework consisting of the following conditions: (1) the existence of NDR is general in the Li-O₂ battery system as shown in Fig. S8 and (2) the NDR potential region was the transition region of the proportional contribution of the solution and surface pathways. There is no change in the framework consisting of the conditions (1) and (2). Here, we have demonstrated the additional condition: (3) the magnitude of the change in the proportional contribution of the reaction pathways in the NDR potential region is generally classified into three types as shown in the newly revised Fig. 6a. That is, Type-1 with large change in proportional contribution and large NDR, and Types-2 and 3 with small change in proportional contribution and small NDR.

The potentiostatic discharging behavior shown in Figs. 5 and S13 attributes each system to the Types as follows. Both DMSO/CP and DMSO/Au systems are classified into Type-2 and the proportional contribution of the reaction pathways varied in the NDR potential region, while solution pathway is dominant at all potentials. The MeCN/Au system belongs to Type-3 and the surface pathway was dominant at all potentials. It should be noted that the results obtained with Au systems were consistent with the conclusion described in the previous manuscript {ref. 29, Johnson, L. *et al.*, *Nat. Chem.* **6**, 1091-1099 (2014)} in which Au plate or disk were applied as working electrodes. Meanwhile, as described in the previous manuscript, the MeCN/CP system belongs to Type-1 and the proportion of the reaction pathways changed dramatically in the NDR potential region. For details and discussions on this framework, please see the corresponding parts highlighted in the newly revised manuscript and Supplementary Information.

The Reviewer's Comment: *Unfortunately, the authors have not responded to my single request (see below) and therefore I cannot support publication;*

... when describing the galvanostatic oscillation (that seen in a battery) the negative shift is said to be due to surface adsorption and the positive shift "can be explained in the exactly same way. Namely, the positive potential shift leads to decrease in the coverage of LiO_2^ and to increase in the ORR current, leading to the further positive shift of the potential." This explanation is not strong and is contradictory*

For clarity, the current oscillation mechanism is clear and the influence of the IR drop logical. But during galvanostatic measurements, no argument has been presented to explain why the "positive feedback", resulting in a negative shift in potential, stops and shifts in the opposite positive direction. My expectation is that this will simply continue to shift negative and reduce solvent. The authors state "The rapid change of the potential from the negative- to positive-end of the potential oscillation can be explained in exactly the same way." This is not sufficient and needs further explanation.

They only state that

"the positive potential shift leads to a decrease in coverage by LiO_2^ and to an increase in the ORR current, producing an additional positive shift of the potential.*

However above state that LiO_2 coverage results in a negative shift.

"in coverage by the LiO_2^ (i.e., the inhibitor) and so decreases the ORR current. This, in turn, results in a further negative shift of the potential, in a positive-feedback manner, to maintain a constant current."*

In summary, there has to be a trigger to drive the positive shift and I do not believe surface coverage is the reason. If the authors are unable to explain this then I would have to conclude that their theory is incorrect.

Response: We appreciate the Reviewer's comment on this point. As the Reviewer noted, the description of the potential shift from negative to positive was simplified in the previous manuscript. In consideration of the fact that NDR and electrochemical oscillation are not familiar to many researchers in the battery research field, we have carefully explained the potential shift from negative to positive in the section titled "Mechanism of the potential and current oscillation" in the newly revised Supplementary Information. The revised parts have been highlighted.

Prior to explaining the revised parts in detail, the relation established between the potential U (and also E), the coverage of LiO_2^* and ORR current value should be clarified. As we described in the previous Supplementary Information, the inhibiting effect of LiO_2^* plays a key role in this system. In the NDR potential region, the negative (or positive) shift in potential leads the increase (or decrease) in the coverage of LiO_2^* (i.e., the inhibitor). This, in turn, results in the decrease (or increase) in the ORR current value, which induces the further negative (or positive) shift in the potential to maintain a constant current. This positive feedback mechanism is the origin of the potential oscillation phenomena in the galvanostatic discharging.

Based on the above relation between the potential, the coverage of LiO_2^* and the ORR current value in the NDR potential region, the potential shift from positive to negative in the potential oscillation phenomena has been explained in the revised version of Supplementary Information, as follows (p. S6, lines 14-20):

"At higher values of U (and also of E), the inhibiting effect is low, as coverage by the LiO_2^* is lower, and therefore the ORR proceeds efficiently. However, a high ORR rate will decrease the surface concentration of oxygen and thus causes a gradual negative shift in the potential so as to maintain a constant current density in a galvanostatic mode. In the NDR region, this negative potential shift leads to increases in coverage by the LiO_2^* (i.e., the inhibitor) and so decreases the ORR current. This, in turn, results in a further negative shift of the potential, in a positive-feedback manner, to maintain a constant current."

Upon this resubmission, the potential shift from negative to positive was more carefully described as follows corresponding to the explanation on the shift from positive to negative described above. The contrasting expressions between the above and below explanations were underlined (p. S6, lines 23-29):

“At lower values of U (and also of E), the inhibiting effect is large, as coverage by the LiO_2^* is higher, and therefore the ORR cannot proceed efficiently. However, a lower ORR rate will increase the surface concentration of oxygen and thus causes a gradual positive shift in the potential so as to maintain a constant current density in a galvanostatic mode. In the NDR region, this positive potential shift leads to decrease in coverage by the LiO_2^* (i.e., the inhibitor) and so increase the ORR current. This, in turn, results in a further positive shift of the potential, in a positive-feedback manner, to maintain a constant current.”

Finally, a general description on NDR and oscillatory phenomena has been shown. An electric circuit having NDR can generally cause an oscillation phenomenon. An oscillation circuit of the Esaki diode is one of the good examples which show electric oscillation caused by NDR. Oscillation phenomena in electrochemical systems associated with NDR is also an example of such electric circuit oscillation and is a research field that have been studied for a long time as summarized in some reviews. It has been now mathematically clarified that the existence of NDR is a required condition for the potential and current oscillation to be occurred. It should be noted here that the current and potential oscillations can appear in an electrochemical system with NDR, irrespective of the molecular origin of the NDR.

We would like to put emphasis here that the description on the potential shifts in our manuscript is based on the electrochemical oscillation mechanism which has already been generalized and established. In the newly revised Supplementary Information, we have also made an additional explanation that this theory already well established, citing the relevant representative literatures. The revised parts were highlighted.

Thank you again for your comments on our paper. We trust that the revised manuscript is now suitable for publication.

RESPONSE TO REVIEWER 3:

Introductory Response: We wish to express our great appreciation to the Reviewer for his or her insightful comments, which have helped us significantly improve the paper. In advance of the response to the comment, we would like to mention the major point of this revision.

The experimental results added to the revised manuscript unveiled the common framework to be applied to Li-O₂ batteries. Meanwhile, we maintain our argument and conclusion which have been discussed in our previous version of manuscripts. The framework, which was additionally shown in the newly revised manuscript, clearly explains both of our results and previous reports in a comprehensive manner. In order to clarify the framework upon this resubmission, additional experiments employing Au electrode were performed and the whole of the manuscript was revised. The outline of the framework is described as follows.

In the previous version of manuscript, we clarified the framework consisting of the following conditions: (1) the existence of NDR is general in the Li-O₂ battery system as shown in Fig. S8 and (2) the NDR potential region was the transition region of the proportional contribution of the solution and surface pathways. There is no change in the framework consisting of the conditions (1) and (2). Here, we have demonstrated the additional condition: (3) the magnitude of the change in the proportional contribution of the reaction pathways in the NDR potential region is generally classified into three types as shown in the newly revised Fig. 6a. That is, Type-1 with large change in proportional contribution and large NDR, and Types-2 and 3 with small change in proportional contribution and small NDR.

The potentiostatic discharging behavior shown in Figs. 5 and S13 attributes each system to the Types as follows. Both DMSO/CP and DMSO/Au systems are classified into Type-2 and the proportional contribution of the reaction pathways varied in the NDR potential region, while solution pathway is dominant at all potentials. The MeCN/Au system belongs to Type-3 and the surface pathway was dominant at all potentials. It should be noted that the results obtained with Au systems were consistent with the conclusion described in the previous manuscript {ref. 29, Johnson, L. *et al.*, *Nat. Chem.* **6**, 1091-1099 (2014)} in which Au plate or disk were applied as working electrodes. Meanwhile, as described in the previous manuscript, the MeCN/CP system belongs to Type-1 and the proportion of the reaction pathways changed dramatically in the NDR potential region. For details and discussions on this framework, please see the corresponding parts highlighted in the newly revised manuscript and Supplementary Information.

The Reviewer's Comment: *My major concern with this study has to do with the conclusion that the NDR represents a boundary region separating surface from solution phase-mediated Li₂O₂ electrodeposition. My critique was that in DMSO, NDR seems to have little effect: the discharge capacities obtained below and above the NDR are quite similar, and no voltage oscillations characteristic of the NDR are obtained in discharge experiments in DMSO.*

The authors respond by stressing that: 1. Figure S8b and d demonstrate an NDR region where the absolute value of the current increases within the capacitive region while the potential is swept in a more positive direction and more importantly, that 2. the NDR more directly describes the adsorption of the ORR inhibitor, but not the extent of coverage given by the adsorption isotherm. Thus, in DMSO, there is a much smoother transition between the surface and solution pathways, leading to the more gradual increase in capacity shown in Figure 5d for DMSO. This qualification is important because although several questions about the NDR remain unresolved (its relative absence from previous Li-O₂ discharge studies with similar experimental setups, the identity of the inhibitor, why no oscillations are observed in DMSO, why a solution-mediated mechanism is proposed to occur at as low a potential as 2.0 V in MeCN, in contrast to other literature) it better rationalizes how observations in MeCN and DMSO belong within the same conceptual framework.

Response: We greatly appreciate the Reviewer's comment. As the Reviewer noted, in the previous manuscript, we regrettably did not clarify a common conceptual framework to be applied to Li-O₂ battery system. In accordance with the Reviewer's comment, we have showed the framework as well as the additionally performed experimental results and the responses to the points noted by the Reviewer. Please see the relevant description of the framework described in the introductory response part and the attached Figure in the final page of this response. Meanwhile, we believe that most of the discussion pointed out by the Reviewer had been evaluated in the previous revision. We would greatly appreciate it if the Reviewer could read again the arguments in the previous manuscript and response reports.

In the Reviewer's Comment, the Reviewer cited the following points which were still unsolved.

Point 1: *in DMSO, NDR seems to have little effect: the discharge capacities obtained below and above the NDR are quite similar*

Point 2: *no voltage oscillations characteristic of the NDR are obtained in discharge experiments in DMSO*

Point 3: *its relative absence from previous Li-O₂ discharge studies with similar experimental setups*

Point 4: *the identity of the inhibitor*

Point 5: *why no oscillations are observed in DMSO*

Point 6: *why a solution-mediated mechanism is proposed to occur at as low a potential as 2.0 V in MeCN, in contrast to other literature*

Our responses to the above points are as follows.

Response to the Point 1: This point can be explained in the above framework described in the introductory response section. As shown in the attached Figure in the final page of this response, DMSO/CP system belongs to the Type-2, of which the proportion of the reaction pathways varies in the NDR potential region while the solution pathway is dominant at all potentials.

Response to the Point 2: As shown in Fig. S8 and mentioned in the section titled “negative differential resistance” in the manuscript, NDR is a universal phenomenon appearing in the ORR of Li-O₂ battery. Meanwhile, as mentioned in the last paragraph of the section titled “Possible origin of NDR and correlation with the reaction pathway”, NDR is not a sufficient but a required condition for electrochemical oscillation. A positive feedback mechanism that destabilizes the system is necessary for the appearance of electrochemical oscillation phenomena. Importantly, the larger NDR appears, the more effective positive feedback mechanism works, eventually inducing electrochemical oscillation. Actually, many examples are reported in which the electrochemical oscillation did not appear when the NDR was small. The relationship between the existence of the NDR and the appearance of oscillation was not only shown by experimental results but also proved by mathematical theory {Koper, M.T.M., *Journal of Electroanalytical Chemistry* **409**, 175 (1996), Koper, M.T.M., *Journal of the Chemical Society, Faraday Transactions* **94**, 1369 (1998) and Strasser, P. et al., *The Journal of Chemical Physics* **107**, 979 (1997)}.

We understand that the framework, which is about the oscillation and NDR in the electrochemistry, should be discussed with due consideration that researchers in the battery research field are unfamiliar. Therefore, we have added a more careful explanation to the section titled

“Mechanism of the Potential and Current Oscillation” in the newly revised Supplementary Information, citing related references.

Response to the Point 3: We showed experimental results obtained with carbon paper and various electrolyte solvents (Fig. 1b), carbon sheets prepared from a porous carbon material (Fig. 1d), and Au-mesh (Figs. S8c and S8d) in the previous revision. We also evaluated the effect of electrolyte (Fig. S2). Moreover, we obtained the results of discharging to show that the discharge behavior was independent of cell structures and the solid-state Li ion-conductive glass-ceramic film (LIC-GC) used as a separator (Fig. S5). Further, the potentiostatic discharging tests with Au-mesh cathode were additionally performed on this revision (Fig. S13 and the attached Figure of this response), therefore, we believe that the experiments corresponding to the results reported previously have been thoroughly verified.

Response to the Point 4: As the reviewer noted, we mentioned that had not identified the inhibitor by electrochemical or spectroscopic methods in the previously submitted manuscript. We hope to address this aspect of the work in future. Meanwhile, we also revealed that it was derived that the inhibitor is LiO_2^* from a logical analysis of the several previous reports. We would like to note here that the above argument was already discussed in the previously submitted manuscript (p. 8, line 34 – p. 9, line 4) and the introductory response section in the previous version of the response to the reviewers.

Response to the Point 5: This noted point is a same as the point 2. Please see the Response to the Point 2.

Response to the Point 6: As we described above, MeCN/CP and MeCN/Au systems belong to the different types, Type-1 and Type-3, respectively. What we can conclude from the appended framework is that the cathode materials essentially affected on the discharge behavior of each system. Moreover, there are several reports in which high discharge capacities were obtained when carbon cathodes were used in MeCN systems {ref. 39, Kwabi, D.G., *et al.*, *Phys. Chem. Chem. Phys.* **18**, 23933-24953 (2016) and McCloskey, B.D., *et al.*, *J. Phys. Chem. Lett.* **3**, 3043-3047 (2012)}. These results in the previous reports were in agreement with our conclusion that solution pathway is dominant in MeCN system under an appropriate conditions even though MeCN is a solvent with low donor number.

Attached Figure| (a and b), The schematic illustration of the changes of ratio in $\text{LiO}_2^*/\text{LiO}_2^s$ (a) and cathodic current (b) with potentials. The NDR potential region indicates the transition region of the ratio ($\text{LiO}_2^*/\text{LiO}_2^s$) in each system. (c and d), Discharge capacities of potentiostatic discharging at various potentials obtained with CP (c) and Au-mesh cathodes (d). The red circles and blue squares represent the results obtained with MeCN and DMSO electrolyte, respectively. The dashed lines indicate the theoretical capacity via the surface pathway. Please see the detailed information described in Fig. 5 (c) and Fig. S13 (d) and related description in the manuscript.

Thank you again for your comments on our paper. We trust that the revised manuscript is now suitable for publication.

Reviewers' comments:

Reviewer #2 (Remarks to the Author):

While I am grateful that the authors have now tried to answer my previous question, unfortunately I still do not believe that the explanation as written is valid. I can accept the authors suggestion that NDR is an indication for a switching mechanism, but only if the NDR can be explained in the context of the lithium O₂ chemistry. Briefly, the authors invoke changing ORR rate, yet the rate (current) is constant in this method. This must be resolved. Perhaps the authors have a valid explanation for this process (I can think of a few options), but thus far this has not been suitably communicated, which lowers my confidence in theory.

The molecular origin of the NDR and the proposed transition from solution to surface pathway are the critical aspect of this paper and must be explained. If not, then the paper is not suitable for Nature Communications.

Reviewer #3 (Remarks to the Author):

I am satisfied with the authors' responses and recommend this work for publication

RESPONSE TO REVIEWER 2:

We wish to express our great appreciation to the Reviewer for his or her insightful comments, which have helped us significantly improve the paper.

The Reviewer's Comment: *While I am grateful that the authors have now tried to answer my previous question, unfortunately I still do not believe that the explanation as written is valid. I can accept the authors suggestion that NDR is an indication for a switching mechanism, but only if the NDR can be explained in the context of the lithium O₂ chemistry. Briefly, the authors invoke changing ORR rate, yet the rate (current) is constant in this method. This must be resolved. Perhaps the authors have a valid explanation for this process (I can think of a few options), but thus far this has not been suitably communicated, which lowers my confidence in theory.*

The molecular origin of the NDR and the proposed transition from solution to surface pathway are the critical aspect of this paper and must be explained. If not, then the paper is not suitable for Nature Communications.

Response: We appreciate the Reviewer's comment on this point. As the Reviewer noted, the description of the requirement for the potential oscillation appearance which we described in the previous version of the Supplementary Information was insufficient. We reviewed carefully the previous version and have revised accurately the description of the mechanism of the appearance of potential and current oscillation. The revised part was highlighted. The detail of this revision is shown below.

According to the general theory of electrochemical oscillation (ref. 48, S10 and S14), existence of negative differential resistance (NDR) and presence of a Faraday current component to hide NDR are necessary for the appearance of potential oscillation. Therefore, potential oscillation phenomena appear with an oscillatory period corresponding to the time scale of the scan rate at which NDR on voltamograms is hidden. In the research field of electrochemical oscillation, the above mentioned potential oscillation is called "hidden-NDR type", and its mathematical model is established (see references mentioned above). In the system of this manuscript, the inhibition of the oxygen reduction reaction (ORR) current by LiO₂* is the origin of NDR and Faraday current hiding the NDR is most probably attributed to the decomposition of the solvent in consideration of the result shown in Fig.

S15. Based on the above theory, the constant current value can be held even if the ORR current value fluctuates in the NDR potential region.

Finally, we have shown our conclusion regarding the molecular origin of NDR. As we described in the previous response letter, the above theory of current and potential oscillations is established, irrespective of the molecular origin of NDR. Meanwhile, as the Reviewer noted, we understand the importance of the identification of the molecular origin of NDR and address it in future. However, we had already discussed and made a present conclusion of the molecular origin of NDR in the 2nd revision of manuscript and response letter submitted in July. The following description is in accordance with the conclusion argued before. As discussed in the manuscript (p. 9, lines 3-13), we revealed that it was derived that the inhibitor is LiO_2^* from a logical analysis based on the results of electrochemical and spectroscopic experiments (ref. 29 and 37) and DFT calculations (ref. 38), which were reported previously. We believe that the above conclusion, which is that LiO_2^* is the molecular origin of NDR, is the most logical and reasonable at present.

Thank you again for your comments on our paper. We trust that the revised manuscript is now suitable for publication.

REVIEWERS' COMMENTS:

Reviewer #2 (Remarks to the Author):

The edits are suitable and I support publication.